# Strategic Candidacy in Generative AI Arenas

Chris Hays [1]    Rachel Li [1]    Bailey Flanigan [1]    Manish Raghavan [1]

## Abstract

AI arenas, which rank generative models from pairwise preferences of users, are a popular method for measuring the relative performance of models in the course of their organic use. Because rankings are computed from noisy preferences, there is a concern that model producers can exploit this randomness by submitting many models (e.g., multiple variants of essentially the same model) and thereby artificially improve the rank of their top models. This can lead to degradations in the quality, and therefore the usefulness, of the ranking. In this paper, we begin by establishing, both theoretically and in simulations calibrated to data from the platform Arena (formerly LMArena, Chatbot Arena), conditions under which producers can benefit from submitting clones when their goal is to be ranked highly. We then propose a new mechanism for ranking models from pairwise comparisons, called You-Rank-We-Rank (YRWR). It requires that producers submit rankings over their own models and uses these rankings to correct statistical estimates of model quality. We prove that this mechanism is approximately clone-robust, in the sense that a producer cannot improve their rank much by doing anything other than submitting each of their unique models exactly once. Moreover, to the extent that model producers are able to correctly rank their own models, YRWR improves overall ranking accuracy. In further simulations, we show that indeed the mechanism is approximately clone-robust and quantify improvements to ranking accuracy, even under producer misranking.

[1]Massachusetts Institute of Technology, Cambridge, MA. Correspondence to: Chris Hays <johnchrishays@gmail.com>.

*Proceedings of the 43rd International Conference on Machine Learning*, Seoul, South Korea. PMLR 306, 2026. Copyright 2026 by the author(s).

## 1  Introduction

As generative AI models proliferate, there is a need to systematically compare their performance. Such evaluations allow AI users to make informed choices between models, for organizations to make informed procurement decisions, for investors to allocate investments to more promising AI labs, for AI researchers to identify promising model development techniques towards making further progress, and for the research community to focus its efforts on common tasks (Donoho, 2024, 2017). This need has motivated the emergence of *generative AI arenas*, in which people vote on the outputs of pairs of different models. These comparisons are then aggregated into an overall ranking over models in the arena (where higher-ranked models are those that win pairwise comparisons more often). This evaluation approach offers the key advantage (over, e.g., static benchmarks) that evaluation occurs in the course of organic use, and so quality is measured on arguably user-relevant dimensions that might otherwise be difficult to capture.

Accordingly, since 2024, many such arenas have emerged (Chiang et al., 2024; Zhao et al., 2025; Chi et al., 2025; Miroyan et al., 2025; Jiang et al., 2024): foremost among them currently is Arena (formerly LMArena, Chatbot Arena) (Chiang et al., 2024), which we will use as our prototypical example. Although these platforms are new, there is already evidence that their rankings are substantively important: consumers (Morrison, 2024), investors (Glover, 2025), and tech companies themselves (Alibaba Cloud Community, 2025) consider generative AI arenas to be a credible and economically relevant signal about the quality of models.

As these arenas' rankings become more important, so grow incentives for companies to try to improve the standing of their models. In recent work, Singh et al. (2025) highlight the risks of strategic manipulations: They submit several identical copies (which we'll call *clones*) of the same models to Arena, finding that one is ranked several places above the other.[1] Intuitively, conditions on these platforms may

---

[1]Arena publishes confidence intervals around estimates of model qualities, and the confidence intervals for the estimated scores for the models in the Singh et al. (2025) experiment were overlapping. Such uncertainty quantification strategies would be useful for assigning ties in the ranks between models. However,

be favorable to strategic candidacy: rankings are formed through a relatively small number of comparisons (typically tens to hundreds per pair of models on Arena, or thousands to tens of thousands of comparisons per model) and pairwise win margins between models are small (for example, at the time of writing, the 1st-ranked model has a 58% win rate against the 20th-ranked model). In this low-data, tightly competitive regime, small amounts of noise can substantially affect outcomes. Thus, model producers have both the incentive and the ability to improve their arena position by simply submitting identical or near-identical[2] models. With this as a starting point, we study two central questions:

1. When can producers benefit from clones in existing ranking systems, and if they can, by how much?

2. Can we design clone-robust ranking systems for this setting?

**Our approach and contributions.** We begin with a formal model of the generative AI arenas. We assume that voter preferences are generated via the statistical model used for inference in these arenas (Chiang et al., 2024), a Bradley-Terry (BT) model. Thus, voter behavior is in some sense "ideal" and the status quo statistical model used for ranking is not misspecified. However, because the number of voter preferences are limited, there is noise in the rankings generated from these preferences. We then analyze the choices of model producers, who we assume have an exogenous set of distinct models (where distinctness of models is not observed by the platform) and may submit multiple copies of the same model to the ranking mechanism. Key to our formulation is the idea that producers may hope to increase the ranking of their top models by exploiting noise in voter preferences.[3] In our formalization of model producer incentives, producers derive utility from being ranked first within various subsets of models ("leaderboards"), like all models, all open-weight models, models below a given cost

or latency threshold, or other intrinsic features of models on which a model producer wishes to compete. We make two primary contributions, corresponding to the research questions above:

*Establishing clone-nonrobustness of status quo mechanisms.* In Section 3, we formally prove conditions under which the intuition above is correct: the status-quo mechanism — which fits BT via maximum likelihood estimation and directly reports the implied ranking — indeed rewards producers for submitting model clones. Each clone effectively gives the producer an extra chance of getting "lucky" with their ranking position. We confirm that this problem is significant in simulations calibrated to Arena data: we show that some models can substantially improve their rank by submitting one or a few clones to the mechanism. To our knowledge, ours is the first proof of clone-nonrobustness of a Bradley-Terry models under a correctly specified, homogeneous human preference model. Existing non-robustness results in the literature hold in circumstances when voters have heterogeneous preferences, so the Bradley-Terry model is misspecified (Procaccia et al., 2025).

*Proposing a new accuracy-improving mechanism, You-Rank-We-Rank (YRWR), with clone-robustness guarantees.* In Section 4, we propose a new mechanism, which we prove is $O(1/\sqrt{s})$-strategyproof, where $s$ is the total number of samples per pairwise comparison. That is, a producer cannot gain more than $O(1/\sqrt{s})$ utility from doing anything other then submitting exactly one copy of every model in their set of distinct models. Our mechanism asks each producer to submit a ranking over their own models. We show that this precisely negates the potential benefits from adding a clone. We also establish that, if producers truthfully report rankings, the mechanism *only makes the ranking more accurate*, and provide conditions under which the mechanism is truthful. Our semisynthetic experiments confirm that YRWR is both clone-robust and increases overall ranking accuracy, even when producers do not necessarily report the correct ranking over their own models. We give an overview of our mechanism versus the status quo in Figure 1.

**Related work.** There is a growing literature on clone-robustness of Bradley-Terry models (Procaccia et al., 2025; Gölz et al., 2025; Siththaranjan et al., 2023). These works focus on a setting where voters may have heterogeneous preferences: different types of voters might have different preference orderings over candidates. Our work is complementary to these in that we explore clone nonrobustness *even when voter preferences are homogeneous*, so that in infinite data, there would be one "correct" ranking over models. In other words, their work studies clone nonrobustness stemming from misspecification of the BT model, and our work studies clone nonrobustness stemming from

---

in this work, we consider the (perhaps more difficult) problem of model ranking mechanisms where ties are not allowed, and indeed, at the time of writing, Arena does not allow ties in the ranks it assigns to models.

[2]For example, near-identical submissions might occur if a producer submits multiple, qualitatively similar checkpoints from the same training run.

[3]This is related to but different from the concerns around "hand picked" scores highlighted in Singh et al. (2025). In their work, the focus is on using private testing (where preference data is collected before a model is included in a ranking) to try to exploit selection effects by releasing only the model ranked best during private testing. There is no such notion of private testing in our work: all models submitted to the mechanism are assumed to be released publicly. Instead, in our model, the producers' incentive to submit multiple models comes from the fact that their utility is determined by their *best-performing* models, rather than the average performance of their models.

finite-sample effects.

Our proposed mechanism is similar to that of Su (2021); Su et al. (2025) implemented at ICML, which consider incentives in the academic peer review process: both involve asking participants to rank their own submissions to the mechanism. However, we incorporate the ranking differently: Whenever the ranking induced by fitted Bradley-Terry scores disagrees with the producer's own ranking, our mechanism resolves disagreement by assigning all of the models in question the minimum of their scores as opposed to fitting an isotonic regression (which effectively takes the mean of scores when reviewer scores disagree with author rankings). In their setting, the self-ranking helps denoise scores for a fixed, exogenous set of candidates; in ours, it also enables clone-robustness when the set of candidates is endogenous — participants in the mechanism can choose which (and how many) candidates to submit. Our setting is also different because we have to account for dependencies among fitted scores (because votes are over *pairs* of models), whereas there is no such dependency in their model. One reason for the differences between their settings and ours is that their motivating applications are reinforcement learning from human feedback (RLHF) (i.e., evaluating model responses for a given prompt against each other) and ours are model ranking (i.e., evaluating models against each other).

Our work contributes to a growing body of work in strategic behavior in AI evaluations. In particular, Chen et al. (2026) analyze a Stackleberg game between a benchmark producer and multiple model producers, where model producers may try to game the benchmark by making benchmark-specific improvements to their model. Another line of work explores strategic or adversarial voting on generative AI arenas (Huang et al., 2025a,b; Min et al., 2025). More broadly, our work sits in a growing literature on strategy-robust statistics (Spiess, 2025; Bates et al., 2024; Shi et al., 2025), which treat statistical protocols as mechanism design problems.

## 2 Setup

There are $n$ model producers. Each producer $i \in [n]$ has a set $\mathcal{K}_i$ of distinct models where $k_i = |\mathcal{K}_i|$ is a constant. The full set of distinct models will be denoted $\mathcal{K} = \bigcup_{i \in [n]} \mathcal{K}_i$ where $k = |\mathcal{K}|$.

**Producer actions.** Each producer $i$ decides how many copies of each distinct model $j \in \mathcal{K}_i$ to submit to the mechanism. In particular, they may submit clones of the same model to try to benefit from evaluation noise. Formally, a producer $i$'s action space is $z_i \in \mathbb{Z}_{\geq 0}^{k_i}$, where $z_{ij}$ denotes the number of copies of model $j$ submitted by producer $i$. We use $z_{-i}$ to represent the actions of all players be-

sides $i$, and $z_{-(i,j)}$ to represent the actions corresponding to all models except producer $i$'s distinct model $j$. We write $z = (z_1, \ldots, z_n)$ to denote a full assignment of actions to producers. The full set of models submitted by producer $i$ to the mechanism is

$$\mathcal{M}_i(z_i) = \bigcup_{j \in \mathcal{K}_i} \{j^{(1)}, \ldots, j^{(z_{ij})}\},$$

and we write $m_i(z_i) = |\mathcal{M}_i(z_i)|$. The full set of models submitted to the mechanism by all producers is $\mathcal{M}(z) = \bigcup_{i \in [n]} \mathcal{M}_i(z_i)$, and we write $m(z) = \sum_{i \in [n]} m_i(z_i)$. When it is clear from context or when we are speaking about a generic set of models, we will drop the argument $z$.

**Pairwise voting.** After the producers submit their models for an action $z$, the users *vote* on pairwise comparisons. After seeing outputs from two anonymous models $j, j' \in \mathcal{M}(z)$, the user answers either $j > j'$ ($j$ is preferred) or $j' > j$. Following the classic Bradley-Terry (BT) model (Bradley & Terry, 1952), we assume each model $j \in \mathcal{M}$ has some latent quality $R_j \in \mathbb{R}_{\geq 0}$, and each vote is drawn as an independent Bernoulli random variable with probability

$$\Pr(j > j') = \frac{\exp R_j}{\exp R_j + \exp R_{j'}}.$$

(Of course, if two models $j, j'$ are clones, then $R_j = R_{j'}$.) We use $p_{j > j'} = \Pr(j > j')$ as shorthand. We use $R$ to denote the vector of qualities, and $R_{(\ell)}$ to denote the $\ell$-th largest value in $R$ (and we will use analogous notation for estimated qualities as well). This voter preference model is exactly that for which Arena's ranking procedure is specified correctly. Our analysis is thus deliberately favorable to the status quo.

Let there be $s \in \mathbb{N}$ votes per pairwise model comparison. We use $v_{j > j'}$ to denote the (random) number of votes in which $j > j'$ (and thus, $v_{j' > j} + v_{j > j'} = s$.) We'll write $v = \{v_{j' > j}\}_{j' \neq j}$ to denote the full set of vote counts. The parameters describing model producers are the number of producers $n$, distinct models $\mathcal{K}$, model qualities $R$ and pairwise vote counts $s$. Together, we will refer to fixed values of these parameters as a *problem instance*.

Throughout this paper, we will impose the following regularity condition on $R$.

**Assumption 2.1.** There exists a universal constant $C$ such that, for all problem instances and $j, j' \in \mathcal{K}$, it holds $|R_j - R_{j'}| \leq C$.

The assumption says that the qualities of any two producers may not be arbitrarily different, and that this maximum difference does not grow even when we analyze problem instances with many distinct models (e.g., large $\mathcal{K}$). We'll

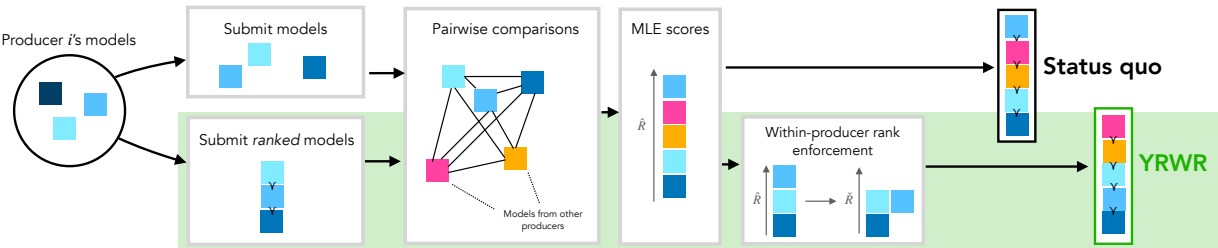

*Figure 1.* The *Status Quo* (SQ) mechanism (top half) and the *You-Rank-We-Rank* (YRWR) mechanism (bottom half).

assume that estimated $\hat{R}$ obeys the same constraint. This kind of condition is standard in analyses of Bradley-Terry models (cf. Simons & Yao (1999)) and is useful in our analysis because it ensures that, when the number of models is large, no one model can have too much influence over the rankings of other models. In practice, pairwise win rates are typically bounded away from 0 and 1, even when the pair of models are ranked far apart.

**Mechanism.** A mechanism G is a mapping from models $\mathcal{M}$ and votes $v$ over pairs of models $j, j' \in \mathcal{M}$ to a ranking $\sigma$ over all models in $\mathcal{M}$. We denote mechanism outputs as $G(\mathcal{M}(z); v)$, where we drop the $v$ when it is clear from context. We let $\Sigma(G(\mathcal{M}(z))$ be the distribution of ranks (over the randomness in votes) induced from running G on an instance where players play $z$.

We will write $\sigma(\ell)$ to indicate the model ranked $\ell$-th in the ranking and $\sigma^{-1}(j)$ to indicate the ranking of model $j$. We use the preference relation $>_\sigma$ to indicate a comparison according to the ranking $\sigma$. Finally, we will often talk about rankings over subsets of models, ordered according to a reward vector. For a generic reward vector $r \in \mathbb{R}^m$ and a subset of models $S \subseteq \mathcal{M}$, we define the ranking over $S$ according to $r$ as

$$\text{rank}(S, r) = \sigma : \sigma_j > \sigma_{j'} \iff r_j \geqslant r_{j'}.$$

**Utilities and leaderboards.** Finally, we must formalize producer utilities. A natural goal for producers, intuitively, is to have one of their models ranked first among all models in the arena. However, treating winning overall as the sole source of utility fails to explain behavior in practice, where producers routinely submit models that will not rank highly, even against their own existing models.[4] Instead, we propose a utility model that rewards producers for being best in *a given class of relevant models*, even if they do not rank first in the overall arena.

---

[4]For example, in August 2025, OpenAI released (and eventually submitted to Arena) several open-weight models (the `oss` series) despite the fact that their performance was clearly limited compared to their existing flagship (closed) models; see https://openai.com/index/introducing-gpt-oss/.

Formally, we consider a collection of *leaderboards* $\mathcal{L} \subseteq \{0, 1\}^{\mathcal{M}}$, where each leaderboard $L \in \mathcal{L}$ is a set of competing models. For example, producers might be interested in ranking first among open-weight models, in which case the relevant leaderboard $L$ would consist of $\{j \in \mathcal{M} : j$ is open-weight$\}$. Similarly, we might consider leaderboards for non-reasoning models, models under a certain size/cost/latency threshold, or models supported by a particular IDE. Leaderboards need not be disjoint. Intuitively, they capture the idea that different consumers have different requirements, and indeed we could microfound this with a consumer choice model where each leaderboard is a consumer's consideration set. Under this utility model, producers may would be incentivised to submit "weak" models that have no chance of ranking first in the overall leaderboard if these models have a chance of winning on some other relevant leaderboard.

With this, we can formally define producer utility. Each producer $i$ assigns some importance $\nu_i(L) \in [0, 1]$ to each leaderboard $L$, which corresponds to the utility $i$ receives for having the top-ranked model in $L$ — that is, for each leaderboard $L \in \mathcal{L}$, producer $i$ gets utility $\nu_i(L)$ for winning $L$ and 0 otherwise. We normalize these utilities so that $\sum_{L \in \mathcal{L}} \nu_i(L) = 1$. Formally, we let $\sigma_L$ be the subranking of $\sigma$ over the models in leaderboard $L$; i.e., for all $j, j' \in L, j >_\sigma j' \iff j >_{\sigma_L} j'$. Then, if the overall ranking from the mechanism is $\sigma$, producer $i$'s utility is

$$u_i(\sigma) = \sum_{j \in \mathcal{M}_i} \sum_{L \in \mathcal{L}} \nu_i(L) \cdot \mathbf{1}(\sigma_L(1) = j).$$

Naturally, clones must belong to the same leaderboards, as leaderboard membership is based on a model's fixed characteristics (e.g., size, cost).

Throughout, we use nonasymptotic big-O notation, so that, e.g., for sequences $\{a_i\}_{i=1}^\infty, \{b_i\}_{i=1}^\infty$ we write $b_i = O(a_i)$ if there exists a universal constant $C$ such that $b_i \leqslant Ca_i$ for all $i$.

## 3  The Status Quo Mechanism

The *Status Quo* mechanism (SQ) used by Arena (Chiang et al., 2024) is depicted in the top half of Figure 1 and

**Algorithm 1:** Status Quo (SQ)

**Input:** Models $\mathcal{M}$; pairwise vote counts $v$
Compute

$$\hat{R} \leftarrow \arg \max_{R \in \mathbb{R}^m} \sum_{j \neq j'} v_{j > j'} \log \Big( \frac{\exp(R_j)}{\exp(R_j) + \exp(R_{j'})} \Big)$$

**return** $\sigma = \text{rank}(\mathcal{M}, \hat{R})$ (breaking ties arbitrarily)

is formally defined in Algorithm 1. SQ takes in a set of models present in the arena and the pairwise vote count over them. Using the votes, it fits estimated rewards $\hat{R} = (\hat{R}_1, \ldots, \hat{R}_m)$ via maximum likelihood estimation.[5] Then, it outputs the ranking implied by $\hat{R}$. Ties can be broken arbitrarily. In the text, we will refer to SQ on inputs $\mathcal{M}, v$ as SQ($\mathcal{M}$), dropping the $v$ argument when it is clear from context.

**The effects of clones.** Before we formalize our main result for this section, we identify the key intuition for how cloning can affect the ranking distribution induced by SQ.

*The "Lottery Ticket" Effect.* The cloned model provides an extra chance for the producer to win, since it receives its own random draw of pairwise votes, which we call the *lottery-ticket effect*. The producer benefits from taking the best outcome among these random draws, thereby increasing the probability of winning. This is a direct consequence of the fact that there are a finite number of votes per pair of models, which leads to randomness in the final ranking — if there were infinite comparisons, the ranking would be deterministic and this effect would disappear.

*The "New Competitor" Effect.* Introducing a clone causes a *new-competitor effect*: new pairwise votes must be collected between the cloned model and the existing models. This changes the competitive environment faced by each model. Intuitively, if the clone is very strong, most existing models will lose more of their matchups relative to the counterfactual without the clone; if it is very weak, they win more. These changes propagate to the fitted BradleyTerry scores, potentially increasing or *decreasing* any individual models win probability, adding complexity to our analysis. However, our workhorse lemma, Lemma D.2, establishes that the new competitor effect is small: the change to any individual model's win probability is no more than $O(1/\sqrt{s})$.

Intuitively, a clone is valuable as long as the lottery ticket effect (which is always positive) outweighs the new com-

petitor effect (which can either be positive or negative).

Our main result characterizes when the lottery ticket effect outweighs the potential drawbacks of the new competitor effect. Intuitively, this is when a producers' models have a reasonable (but uncertain) chance of winning a (set of) leaderboard(s) of non-negligible importance. We next formalize these conditions via a definition.

Informally, the definition says that, across leaderboards with positive total weight to a producer $i$, two quantities are bounded away from 0: the model $j$'s chance of winning, and producer $i$'s chance that none of their models win. These conditions are important for a model to be worth cloning because, if a model has no chance of winning, a cloned version of that model also won't have a chance of winning. Similarly, if a producer is guaranteed to win on a set of leaderboards, there is no need to clone models to improve the producer's chances on those leaderboards.

**Definition 3.1** (($\varepsilon, \delta$)-competitive model)**.** For a fixed strategy profile $z$, a model $j \in \mathcal{M}_i$ for producer $i$ is said to be ($\varepsilon, \delta$)-*competitive* if there exists a set of leaderboards $S \subset \mathcal{L}$ such that $\sum_{L \in S} \nu_i(L) \geqslant \varepsilon$ and, for all $L \in S$,

$$\Pr_{\sigma \sim \Sigma(\text{SQ}(\mathcal{M}(z)))} (\sigma_L(1) = j) \geqslant \delta, \quad \text{and}$$

$$\Pr_{\sigma \sim \Sigma(\text{SQ}(\mathcal{M}(z)))} (\sigma_L(1) \notin \mathcal{M}_i) \geqslant \delta.$$

The parameter $\varepsilon$ represents how important the set of leaderboards must be, and $\delta$ represents the minimum probability that model $j$ wins on these leaderboards. The condition that there must exist a set of leaderboards with total utility more than $\varepsilon$ is important because of dependencies across leaderboards: if a clone helps the producer win on an inconsequential leaderboard ($\nu_i(L) \approx 0$) but harms the producer on another leaderboard with non-negligible potential utility, then the costs of cloning might outweigh the benefits.

We are now ready to state our main theorem in this section. It establishes that if a producer has a ($\varepsilon, \delta$)-competitive model, then for a sufficiently large set of models and votes per pair of models, producers are incentivised to submit another copy of that model.

**Theorem 3.2** (Clone-nonrobustness of the status quo mechanism)**.** *For all constants $\varepsilon, \delta > 0$, there exists $s_0, m_0$ such that for all $s \geqslant s_0, m \geqslant m_0$, the following holds. For any producer $i$, any strategy profiles $z$ and any ($\varepsilon, \delta$)-competitive model $j$, producer $i$ would benefit from submitting an additional copy of $j$. Formally, let $z' = (z_{i,j} + 1, z_{-i,j})$. Then*

$$\mathbb{E}_{\sigma \sim \Sigma(\text{SQ}(\mathcal{M}(z')))}[u_i(\sigma)] > \mathbb{E}_{\sigma \sim \Sigma(\text{SQ}(\mathcal{M}(z)))}[u_i(\sigma)].$$

Intuitively, the smaller $\varepsilon$ and $\delta$ are, the weaker the benefits to cloning may be — smaller $\epsilon$ means that the cloned model

---

[5]The Bradley-Terry model parameters are invariant to addition by a constant vector (i.e., data generated by parameters $R$ and $R + c\mathbf{1}$ are equal in distribution), so to fit MLE it is necessary to enforce an identifiability constraint like $\sum_{j \in \mathcal{M}} \hat{R}_j = 0$.

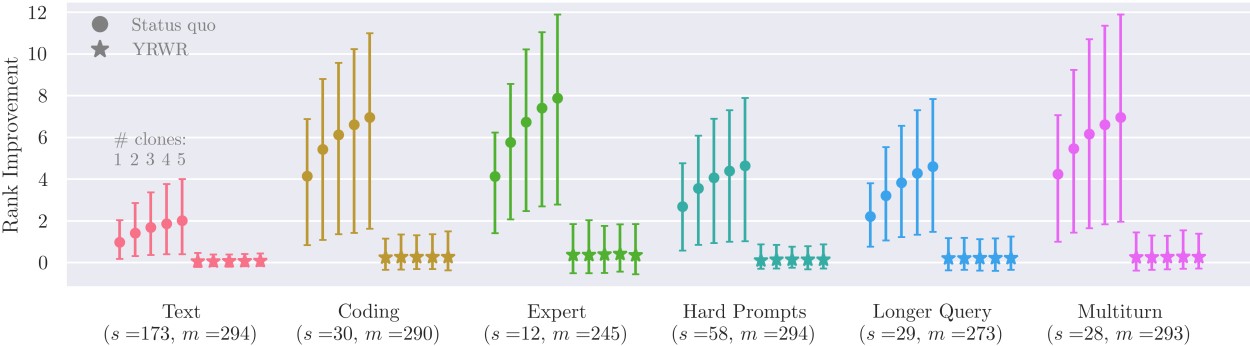

*Figure 2.* Ranks gained via cloning under the *Status Quo* versus *You-Rank-We-Rank* mechanisms, across several of Arena's model arenas.

sits on less important (to the model producer) leaderboards and smaller $\delta$ means that the lottery ticket effect of an additional copy of the $(\epsilon, \delta)$-competitive model are smaller. Thus, smaller $\varepsilon, \delta$ imply larger $s_0, m_0$, since new competitor effects decrease in $s$ and $m$. The proof of Theorem 3.2 is given in Appendix E.

We have thus established conditions under which a producer can benefit from clones. It is trivial to demonstrate that the gains from clones can be potentially very large:

**Example 3.3** (Constant possible gain). Consider $n$ producers with one distinct model, each of identical quality. Each producer wins with probability $1/n$. Now, suppose producer 1 submits $k$ clones (including their original model); their new probability of winning is $k/(k + n - 1)$, which is constant (in $n$ and $m$) for $k \in \Omega(n)$ and approaches 1 as $k \to \infty$.

The fact that the above example sets all rewards identically is for simplicity of intuition; what it illustrates more broadly is that a producer can flood the system with models and drive their win probability upwards arbitrarily.

### 3.1 Simulation study with Arena data

To demonstrate the implications of Theorem 3.2, we conduct a set of simulations calibrated to Arena data to explore how much producers can benefit from clones.

To do this, we snapshot Arena on January 1, 2026 and focus on the largest arenas (those with the most models). For each such arena, we treat all listed models as distinct and use the platform's published BT scores as "ground-truth" qualities, with model producers given by the associated organization metadata. Thus, our empirical approach is designed to show what would happen in the idealized case (favorable to the status quo) in which (1) the Bradley-Terry model was correctly specified and (2) Arena had perfectly estimated model qualities. Holding these qualities fixed, we then vary the number of clones of each model $j$ over

$\ell \in \{1, 2, 3, 4, 5\}$, each time, producing some new synthetic set of models $\mathcal{M}$ with the original models and the clones. For each of these model sets $\mathcal{M}$, we generate synthetic pairwise outcomes: for each unordered pair $(j, j')$, we draw $s$ iid Bernoulli comparisons with BT win probability $p_{j \succ j'}$, where $s$ is the arena's average number of votes per model-pair. This produces our vote counts $v^{(j,\ell)}$ (where $j$ is cloned $\ell$ times). We also repeat this procedure in the raw instance with no clones, the vote counts for which we call $v$. Code to reproduce the simulations and plots is available at https://github.com/johnchrishays/strategic-candidacy-in-genai-arenas.

We then run SQ with vote counts $v^{(j,\ell)}$ for all $j \in L, \ell \in \{1, 2, 3, 4, 5\}$, and also on the raw instance $v$ (using the corresponding multiset of models each time). For each model $j$, we report the highest rank among all of its clones, capturing the idea that producers are rewarded for their highest-ranked model in a given leaderboard. We show the results in Figure 2, where we plot the average, 5th, and 95th percentile of the number of ranks gained across models $j \in L$ between 0 clones (raw instance) and $\ell$ clones, across arenas. The results for SQ are on the left per arena; the right shows analogous results for our mechanism YRWR, which we will unpack in Section 4.4.

Focusing on the lines pertaining to SQ, we see that across arenas, cloning a model leads it to gain in rank position, with some models moving up $\geq 7$ positions with just one clone. With additional clones, rank position can be reliably increased further with mild diminishing marginal returns. We remark that there are larger benefits from clones in arenas (like Coding, Expert, and Multiturn) with fewer pairwise votes per pair of models ($s \leq 30$). Second, we conduct additional data analysis in Appendix B to show that the models that benefit from clones have scores that are clustered together with several other models, which means that small increases to fitted scores yield large increases in position on the arena.

---

**Algorithm 2:** You-Rank-We-Rank (YRWR)

---

**Input:** Models $\mathcal{M}$; rankings $\{\pi_i\}_{i\in[n]}$; vote counts $v$
Compute

$$\widehat{R} \leftarrow \arg\max_{R\in\mathbb{R}^{|\mathcal{M}|}} \sum_{j\neq j'} v_{j>j'} \log\left(\frac{\exp(R_j)}{\exp(R_j)+\exp(R_{j'})}\right)$$

**foreach** *producer* $i \in [n]$ **do**
    **foreach** $j \in \mathcal{M}_i$ **do**
        $\check{R}_j \leftarrow \min\{\widehat{R}_{\pi_i(k)} : k \leqslant \pi_i^{-1}(j)\}$

**Output:** $\sigma = \mathrm{rank}(\mathcal{M}, \check{R})$ (breaking ties within each producer $i$ according to $\pi_i$ and ties between producers arbitrarily).

---

## 4 The You-Rank-We-Rank Mechanism

The YRWR mechanism is depicted in the bottom half of Figure 1 and defined formally in Algorithm 2. It augments the status-quo mechanism first by taking an additional input: a set of *producer-defined rankings* $\pi = \{\pi_i\}_{i\in[n]}$, where $\pi_i$ is a ranking over $\mathcal{M}_i$. Our main result does not require us to assume that this ranking is "correct" in any sense; this can be thought of as a producer's *prioritization* over their models with respect to the mechanism. We will discuss this in more formality later. We will refer to YRWR on inputs $\mathcal{M}, \pi, v$ as $\mathrm{YRWR}(\mathcal{M}, \pi; v)$, again dropping the $v$ from the notation when clear from context.

As in SQ, YRWR first estimates quality scores $\widehat{R}$ via MLE. Then, instead of directly outputting the implied ranking, it performs a *monotone score correction* within each producer: for every model $j \in \mathcal{M}_i$, its corrected score $\check{R}_j$ is the minimum estimated score among all models $j' \in \mathcal{M}_i$ ranked ahead of $j$ according to $\pi_i$. Finally, YRWR outputs a global ranking implied by $\check{R}$.

Notably, our mechanism receives *no external information* about which models are clones and which are distinct. This is motivated by key practical challenges: the platform may have no access to proprietary models' weights or logits and tests of whether models produce similar outputs might be computationally or statistically intractable. Moreover, even if the mechanism could require white-box model inspection, producers might minimally change the parameters of similar models or otherwise manipulate their submissions to evade clone detection. Before we state our main result of this section, we provide intuition about why the self-ranking mechanism provides clone-robustness.

**Intuition: Why do producers' rankings help?** Enforcing that scores obey producer ranks help produce more accurate rankings and disincentivise clones. To see why this is true, consider the distribution of fitted scores of 2 copies of model $j$ by producer $i$, denoted $j^{(1)}, j^{(2)}$, where without

loss of generality, we assume $j^{(1)} >_{\pi_i} j^{(2)}$. Because of the minimum operation to compute $\check{R}_{j^{(1)}}, \check{R}_{j^{(2)}}$, the distribution of their maximum is exactly that of $\widehat{R}_{j^{(1)}}$:

$$\max\{\check{R}_{j^{(1)}}, \check{R}_{j^{(2)}}\} \stackrel{d}{=} \widehat{R}_{j^{(1)}}.$$

Put another way, with YRWR, the producer's fitted score is the maximum of two draws half the time (if $\widehat{R}_{j^{(1)}} > \widehat{R}_{j^{(2)}}$, agreeing with the producer ranking $\pi_i$) and the minimum otherwise. And for two identically distributed random variables, the distribution of a random variable which is the maximum of the two variables with probability half and the minimum with probability half is exactly equal to the distribution of each random variable. This means, from the perspective of the top producer-ranked model in the clones of $j$, it is no better to submit two models than it is to submit a single model.

The key idea we are leveraging is that when submitting clones, the producer cannot know which will do better — the producer will have had to "pick a winner" between the clones in advance. By contrast, under SQ, the distribution of $\max\{\widehat{R}_{j^{(1)}}, \widehat{R}_{j^{(2)}}\}$ stochastically dominates that of $\widehat{R}_{j^{(1)}}$. This creates the selection-on-winners effect which producers could benefit from. Finally, we note that this mechanism removes the incentives for clones despite not having any special information about which models are clones.

### 4.1 Approximate clone-robustness of YRWR.

We now show our main result in this section: for all producers $i$, it is an approximate dominant strategy to submit one copy of each distinct model, *regardless* of producers' submitted rankings $\pi$.

**Theorem 4.1** (Approximate cloneproofness). *For all $\varepsilon > 0$, there exists $s_0, m_0$ such that for all $s \geqslant s_0$ and $m \geqslant m_0$, the following holds. Fix any $\pi, z$, and let $z' = (\mathbf{1}, z_{-i})$ be the profile where $i$ instead plays one copy of each model. Then*

$$\mathbb{E}_{\sigma\sim\Sigma(\mathrm{YRWR}(\mathcal{M}(z'),\pi))}[u_i(\sigma)])$$
$$\geqslant \mathbb{E}_{\sigma\sim\Sigma(\mathrm{YRWR}(\mathcal{M}(z),\pi))}[u_i(\sigma)] - \varepsilon.$$

Our proof of this result is in Appendix F. It relies on the intuition provided above along with an application our workhorse distributional stability result, Lemma D.2: We observe that the distribution of the first mechanism-ranked clone is equal to that of the first producer-ranked clone, by isotonicity. Then, using Lemma D.2, we establish that the win probability of the first producer-ranked clone is approximately (up to additive $\varepsilon$) equal to the distribution of a single model under the strategy with one copy of each model.

## 4.2 Accuracy of YRWR.

Thus far, we have established approximate clone-robustness of the YRWR mechanism. It is natural to ask whether this clone-robustness property comes at a cost to ranking accuracy. After all, our mechanism may modify estimated BT scores, even when there are no clones and when the pairwise preference data is generated by a BT model. Naturally, the impact of producer rankings on the accuracy of the mechanism depends to some degree on the quality of producers' submitted rankings — but we now show that if producers' submitted rankings are consistent with the ground-truth ranking, our mechanism *only makes the scores more accurate*, at least with respect to $\ell_\infty$ distance. We will henceforth refer to the correct ranking as $\pi_i^* := \mathrm{rank}(\mathcal{M}_i, R)$ with $\pi^* := \{\pi_i^*\}_{i \in [n]}$.

**Proposition 4.2** (YRWR is accuracy-improving). *Fix $R, \mathcal{M}$ and let $\hat{R} = \mathrm{SQ}(\mathcal{M})$ and $\check{R} = \mathrm{YRWR}(\mathcal{M}, \pi^*)$. Then,*

$$\|\check{R} - R\|_\infty \leq \|\hat{R} - R\|_\infty.$$

The proof of Proposition 4.2 is in Appendix G and follows from the fact that YRWR's score-correction replaces each score by a minimum over a fixed subset of scores — a mapping that is 1-Lipschitz in $\|\cdot\|_\infty$, so the correction cannot increase $\ell_\infty$ error.

Proposition 4.2 directly implies that that YRWR maintains the asymptotic correctness properties of SQ. We formalize this next.

**Corollary 4.3** (Efficiency and correctness of YRWR). *Fix $R, \mathcal{M}$, and let $\check{R} = \mathrm{YRWR}(\mathcal{M}, \pi^*)$. Then,*

- *$\check{R}$ is a $\sqrt{s}$-consistent estimator of $R$*

- *If $\exists \gamma > 0 : \min_{j \neq j' \in \mathcal{M}} |R_j - R_{j'}| > \gamma > 0$, then*

$$P[\mathrm{rank}(\mathcal{M}, \check{R}) = \sigma^*] \to 1 \ as \ s \to \infty.$$

Informally, the corollary states that, under truthful producer rankings, the modified scores produced by YRWR inherit the statistical efficiency and asymptotic almost sure ranking correctness of the SQ mechanism.

## 4.3 Truthfulness in producer rankings $\pi$.

The accuracy results above rely on producers ranking their models according to $R$. However, it is not immediately obvious that they will always have an incentive or knowledge to do so. In general, misreported producer rankings could lead to reductions in the accuracy of the ranking: Even with infinite data, if a producer misreports their ranking, the mechanism could drastically change the ranking to enforce isotonicity, leading to rankings which are far from correct. We explore producer's ranking strategies and their implications next.

**Incentives for producer misreports.** Even if a producer knows the ground-truth ranking over their own models, they may not be incentivised to truthfully report the ranking. The problem is producers' differential utilities across different leaderboards; we illustrate this with the following example.

**Example 4.4.** Suppose producer $i$ has two models $a, b$ with similar true rewards but where $R_a > R_b$. Suppose $a$ is closed-weight and $b$ is open-weight. Assume the producer knows that they have a negligible chance of winning the leaderboard $L_{\mathrm{all}}$ consisting of all models. I.e., $P(\sigma_{L_{\mathrm{all}}}(1) \in \{a, b\}) \approx 0$. Moreover, suppose the producer assigns non-negligible weight to the leaderboard for open-weight models $L_{\mathrm{open}}$, on which $b$ is eligible but $a$ is not, i.e., $b \in L_{\mathrm{open}}, a \notin L_{\mathrm{open}}$, and $\nu_i(L) \gg 0$. Also, suppose model $b$ has a non-negligible chance of winning $L_{\mathrm{open}}$, i.e., $P(\sigma_{L_{\mathrm{open}}}(1) = b) \gg 0$. Under the YRWR score correction, if the producer reports $a > b$, then $b$'s corrected score is $\check{R}_b = \min\{\hat{R}_a, \hat{R}_b\}$, so a low realized $\hat{R}_a$—which can occur purely due to estimation noise, despite the fact that $R_a > R_b$—will *cap* $b$'s corrected score and reduce $b$'s chance of winning leaderboard $L_{\mathrm{open}}$.

Intuitively, what is happening is that producer $i$ misranks their models to "protect" model $b$ from being penalized due to noisy estimates of lower priority models. This problem is resolved if the qualities of models that *could win* or *could drag down winners* are sufficiently separated relative to $s$ such that the noise poses no risk. While one can formulate sufficient conditions for truthfulness of this flavor, we next state the simpler claim that once $s$ is large enough, YRWR is truthful in $\pi$:

**Proposition 4.5** (Asymptotic truthfulness). *For all $z$ and $\varepsilon > 0$, there exists sufficiently large $s_0 \in \mathbb{N}$ such that for all $s \geq s_0$,*

$$\mathbb{E}_{\sigma \sim \Sigma(\mathrm{YRWR}(\mathcal{M}(z), (\pi_i^*, \pi_{-i})))}[u_i(\sigma)]$$
$$\geq \mathbb{E}_{\sigma \sim \Sigma(\mathrm{YRWR}(\mathcal{M}(z), (\pi_i, \pi_{-i})))}[u_i(\sigma)] - \varepsilon.$$

In other words, truthfulness is an approximately dominant strategy for sufficiently large $s$. Of course, the large $s$ regime is exactly where the YRWR mechanism is least necessary (since in large samples, there are smaller incentives for strategic candidacy).

In Appendix A, we discuss two variants of YRWR, one which is uncertainty-aware (i.e., it overrides producer rankings when enough votes have been collected to confidently order pairs of models) and the second is within-leaderboard (i.e., it only performs score corrections on a leaderboard-by-leaderboard basis, rather than performing a global score correction and using it in all leaderboards).

### 4.4  Simulation study with Arena data

**YRWR vs SQ on rank positions gained (Figure 2).** We now unpack the results on YRWR presented in Figure 2, which are shown on the righthand side for each arena. Our methods are exactly the same as in Section 3.1 except that when testing YRWR, we had to additionally generate each producer $i$'s ranking over their own models $\pi_i$, including any potential clones. For a given set of submitted models produced by producer $i$, in the simulations for Figure 2, we assume the producer ranked them accurately, i.e., as $\pi_i = \mathrm{rank}(\mathcal{M}_i, R)$.

We see a striking difference between the two mechanisms: even on arenas where $s$ is small, YRWR admits *essentially zero* gains in rank for *any* model via cloning — even as the number of clones increases. Nonetheless, there are models that see small rank improvements from submitting clones to the mechanism, as a result of the new competitor effect. In Appendix B, Figure 5, we visualize the benefits of cloning for each model relative to model quality. We show that models near the middle of the ranking and for which there are many models are similar quality are the main beneficiaries of cloning under YRWR, while models with top or bottom true qualities see nearly zero benefits to cloning. We leave further analysis of how the new competitor effect varies with the setting for future work.

**YRWR versus SQ on Accuracy (Figures 3a and 3b)** Our theory tells us that under correct producer rankings $\pi_i$, YRWR should produce more *accurate* reward estimates than SQ (in infinity norm). Unsurprisingly, we find that these more accurate reward estimates translate to more accurate rankings: in Figure 3, we compare the bubble-sort distances (also called Kendall-Tau distance (Kendall, 1938)) from the respective rankings produced by YRWR and SQ to $\mathrm{rank}(\mathcal{M}, R)$, the ground-truth ranking. We see that across all six arenas and all numbers of clones, YRWR has lower distance to the true ranking than SQ, often by many tens of swaps. This is increasingly true as the number of clones increases, illustrating the importance of clone-robustness in recovering accurate rankings.

Our theory leaves open whether these gains in accuracy are robust to *inaccuracies* in producers' rankings, i.e., scenarios in which $\pi_i \neq \mathrm{rank}(\mathcal{M}_i, R)$. We test this by repeating the analysis in with producers' rankings perturbed via a random utility model. To compute perturbed producer rankings $\tilde{\pi}_i$, we add i.i.d Gaussian noise $\epsilon$ to the model qualities:

$$\tilde{R}_j = R_j + \varepsilon$$

We sweep over the variance of $\varepsilon$ such that the Kendall-Tau distance between the perturbed and true rankings is between 10% the length of the list and more than 100% of the length of the list, so that for a ranking of about 300 models,

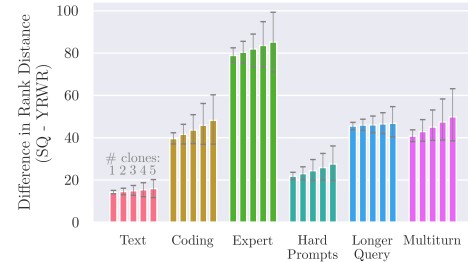

*(a)* Correct producer rankings.

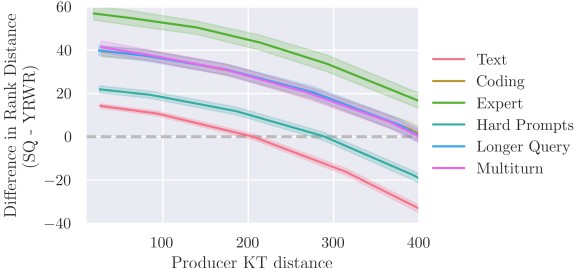

*(b)* Noisy producer rankings.

*Figure 3.* Difference in Kendall-Tau distance to the ground truth under the *Status Quo* versus *You-Rank-We-Rank* mechanisms, across Arena's various arenas. Difference greater than 0 implies that the YRWR mechanism is closer to the true ranking.

the Kendall Tau distance between $\tilde{R}$ and $R$ is between 30 and 400.

We show the results of this analysis in Figure 3b. We see that the ranking by YRWR remains substantially more accurate than SQ for most settings of the noise variance across arenas, with the accuracy advantage diminishing as the noise increases. Eventually, when producer rankings are sufficiently noisy, the accuracy benefits of YRWR drop below zero, although this only happens when the noise in rankings is on the same order as the number of models.

## 5  Discussion

Generative AI arenas serve as important and useful mechanisms to compare AI models under realistic use conditions. Our work studies a simple vulnerability in status quo mechanisms: producers can leverage the noise inherent to these rankings by submitting identical or near-identical models. Such cloning-based manipulations can in turn further deplete samples, leading noise to drown out signal. The alternative mechanism we propose, YRWR, reduces incentives for this type of strategic behavior. As these arenas increasingly guide the development and adoption of AI models, developing mechanisms that are resilient to strategic manipulation is essential to ensuring that rankings remain a trustworthy and accurate signal for the entire community.

## Acknowledgments

The authors thank Nathan Jo, Juanky Perdomo, Jann Spiess, Tijana Zrnic and the participants of the Stanford Causal Inference seminar for helpful discussions and feedback on this work.

## Impact statement

This paper presents work whose goal is to advance the field of machine learning evaluation. There are many potential societal consequences of our work, none of which we feel must be specifically highlighted here.

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

# A   Variants of YRWR

## A.1   An uncertainty-aware YRWR variant

Even if a producer wishes to truthfully report their ranking over models, they may have uncertainty over the relative performance of their models. This could again lead to degradations in the quality of the ranking: If a producer has no information about which models are better than others, their ranking can be no better than random and, even with infinite data, the mechanism would likely produce an incorrect ranking. There are two ways to mitigate incorrect producer rankings due to producer uncertainty.

First, the platform could implement private testing, which allows for the collection of preference data before a model is submitted to public leaderboards. Private testing can serve as a useful tool in circumstances where model producers are uncertain about the relative performances of their models: the producer can collect preference data about the relative performance of their models, and use it to inform the ranking they submit to the mechanism. Indeed, if each producer has only a small number of models, private testing can be very statistically efficient: There are only a small number of pairwise comparisons to make, so high-quality producer rankings can be computed with much less data than necessary for a high-quality overall ranking. Platforms like Arena already provide private testing, and as long as fresh data is collected (i.e., private testing data is not used to form the ranking) when the model is released publicly, model producers cannot benefit from selection effects due to noise in preference data during private testing.

Second, the platform could implement a variant of the YRWR mechanism that only enforces producer rankings among fitted model scores that have overlapping confidence intervals. That is, at confidence level $\alpha/\binom{m}{2}$, let $\widehat{\mathrm{CI}}_{\alpha/\binom{m}{2}}(j)$ be a $\sqrt{s}$-consistent confidence interval for model $j$ containing the MLE estimate $\hat{R}$ (perhaps as computed in Chiang et al. (2024)). Then this *uncertainty-aware* (UA) YRWR variant, called UA-YRWR, would be defined by fitting the BT-MLE scores as in Algorithm 2 but correcting scores as

$$\check{R}_j^{\mathrm{UA}} \leftarrow \min\{\widehat{R}_{\pi_i(k)} \ : \ k \leqslant \pi_i^{-1}(j),$$
$$\widehat{\mathrm{CI}}_\alpha(j) \cap \widehat{\mathrm{CI}}_\alpha(k) \neq \varnothing\},$$

and using these scores to produce a ranking. Like the vanilla YRWR, the mechanism will take $\mathcal{M}, \pi$ as arguments, but it will also take a simultaneous confidence level $\alpha$, which will be assumed to construct a set of confidence intervals that hold simultaneously with probability at least $1 - \alpha$. This variant of the mechanism is appealing because it ignores producer rankings in regimes where there is enough data to confidently rank models. Thus, in infinite data, the ranking produced by this variant would be correct, regardless of the rankings submitted by model producers. To achieve approximate clone-robustness with high probability, the confidence level $\alpha$ would have to be chosen to ensure simultaneous validity across all confidence intervals. We next prove analogous results for the uncertainty-aware variant as we did for the vanilla variant in Theorem 4.1 and Corollary 4.3. Proofs are deferred to Appendix G.

Corollary A.1 (to Theorem 4.1) establishes approximate clone-robustness. Corollary A.1 is identical to Theorem 4.1, except that it is looser by the simultaneous confidence level $\alpha$. The argument for Corollary A.1 is almost directly implied by Theorem 4.1: If the confidence intervals cover $R$, the argument for Theorem 4.1 goes through directly. If not, the change in utility can be at most the confidence level $\alpha$.

**Corollary A.1** (Approximate cloneproofness of UA-YRWR). *For all $\varepsilon > 0$, there exists $s_0, m_0$ such that for all $s \geqslant s_0$ and $m \geqslant m_0$, the following holds. Fix any $\pi, z$, and let $z' = (\mathbf{1}, z_{-i})$ be the profile where $i$ instead plays one copy of each model. For any simultaneous confidence level for UA-YRWR $\alpha > 0$, it holds*

$$\mathbb{E}_{\sigma \sim \Sigma(\text{UA-YRWR}(\mathcal{M}(z'), \pi, \alpha)}[u_i(\sigma)])$$
$$\geqslant \mathbb{E}_{\sigma \sim \Sigma(\text{UA-YRWR}(\mathcal{M}(z), \pi, \alpha))}[u_i(\sigma)] - \varepsilon - \alpha.$$

Proposition A.2 establishes $\sqrt{s}$-consistency of the fitted scores and asymptotic correctness of the estimated ranking. Proposition A.2 is considerably stronger than Corollary 4.3: it holds for any producer ranking, rather than just truthful ones. The argument for the proposition is also different: Since we do not assume the producer ranking is truthful, we cannot appeal to Proposition 4.2, so in the proof we make a direct argument for efficiency and correctness.

**Proposition A.2** (Efficiency and correctness of UA-YRWR). *Fix $R, \mathcal{M}$, and any set of producer rankings $\pi$. Let $\check{R}^{\mathrm{UA}} = $ UA-YRWR$(\mathcal{M}, \pi, \alpha)$. Then,*

- $\check{R}^{\mathrm{UA}}$ *is a* $\sqrt{s}$*-consistent estimator of* $R$

- *If* $\exists \gamma > 0 : \min_{j \neq j' \in \mathcal{M}} |R_j - R_{j'}| > \gamma > 0$*, then*

$$P[\mathrm{rank}(\mathcal{M}, \check{R}^{\mathrm{UA}}) = \sigma^*] \to 1 \ as \ s \to \infty.$$

## A.2   A within-leaderboard YRWR variant

In the results we have presented so far, we have established that YRWR incentivizes truthful reports in the asymptotic regime (Proposition 4.5) but that in general, producers may misreport their true rankings in finite samples (Example 4.4). However, if we modify the mechanism to perform *within-leaderboard* instead of global score correction, model producers are incentivised to submit truthful rankings in finite samples as well, as we show in our next result. Formally, this modified mechanism, called LOCAL-YRWR, is the same as YRWR except that its score correction is performed within each leaderboard:

$$\check{R}^{\mathrm{local}}_{j;\pi,L} := \min\left\{\hat{R}_{\pi_i(k)} : \ k \leqslant \pi_i^{-1}(j) \text{ and } \pi_i(k) \in L\right\}.$$

To implement this approach, leaderboards must be explicitly defined on the platform, rather than implicitly defined by, e.g., a user who will choose the first among a subset of models. That is, the platform must be able to provide separate rankings for different leaderboards. To accomplish this, platforms could provide filters on the rankings within each arena, using metadata about each model, like model size, cost, latency, or other factors. Thus, users who wanted, e.g., to see the ranking among models below a given cost threshold, could see a ranking generated to be both truthful and clone-robust.[6]

Our next result establishes truthfulness of the LOCAL-YRWR mechanism.

**Proposition A.3** (Truthfulness of LOCAL-YRWR). *Fix any producer* $i$*, any strategy profile* $z$ *in which producer* $i$ *submits exactly one copy of each model* $(\mathcal{M}_i = K_i)$*, and fix any other-producer rankings* $\pi_{-i}$*. Then for every finite* $s$ *and every alternative report* $\pi_i$*,*

$$\mathbb{E}_{\sigma \sim \Sigma(\mathrm{LOCAL\text{-}YRWR}(\mathcal{M}(z),(\pi_i^\star,\pi_{-i})))}[u_i(\sigma)] \geqslant$$
$$\mathbb{E}_{\sigma \sim \Sigma(\mathrm{LOCAL\text{-}YRWR}(\mathcal{M}(z),(\pi_i,\pi_{-i})))}[u_i(\sigma)].$$

An analogous approximate strategy-proofness result holds for LOCAL-YRWR as for YRWR, using the same argument as in the proof of Theorem 4.1. Intuitively, it is always approximately utility improving to remove clones from *any* rank correction operating over a set of clone, regardless of which other models by the same producer might or might not be in the same leaderboard.

## B   Additional empirical results

In this section, we provide additional empirical results corresponding to our semisynthetic experiments on Arena data. In Figure 4, we plot the rank improvement attained on average by adding an additional clone. The horizontal axis is the ground truth score in our simulations (i.e., the score assigned by Arena), and the vertical axis is the number of positions the average (across simulations) of the maximum of the ranks of the clones minus the average rank of the model without a clone. The mean rank difference varies widely across models and between leaderboards. There are some models and leaderboards, like Expert, Multiturn and Coding, where cloning a single model can produce a ranking increase of around 8 positions on average. The fact that there may be large benefits to clones on these leaderboards in particular may be related to two factors. First, there are many fewer votes per pair of models on these more specialized leaderboards: each of expert, multiturn and coding have fewer than 30 votes per pair of models on average. Second, the models that benefit from clones in these leaderboards have scores that are clustered together with several other models, which means that small increases to fitted scores yield large increases in position on the leaderboard. By contrast, several leaderboards, like Text, exhibit much smaller benefits to clones. This is because there are relatively more voters per pair of models.

We also note that the benefits of clones mostly disappear for the very best models (furthest to the right on each plot) and the very worst models (furthest to the left on each plot). This may be related to the fact that model qualities are less concentrated at the tails. Also, the very best models can only benefit from clones insofar as they are not ranked first in the

---

[6]However, even if such a filter system could be feasibly implemented, a possible downside of LOCAL-YRWR is that it can lead to inconsistent rankings between pairs of models across leaderboards, which users might find confusing or difficult to interpret.

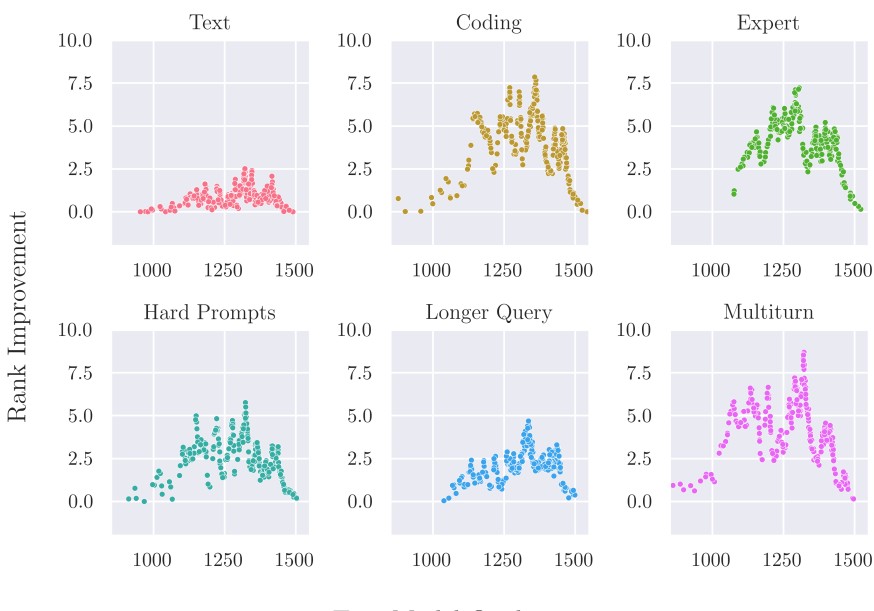

*Figure 4.* Rank difference between submitting one clone and no clones under the SQ mechanism.

simultations without clones, which creates a ceiling for the benefits that can be attained via clones. For example, a model that is never ranked below 3rd place without clones can only improve by up to two positions.

We plot the analogous results for the YRWR mechanism in Figure 5. In each of the panels, the benefits of clones under YRWR average around zero, although there are some models that can see rank improvement of up to around two positions due to the vote reweighting effect. The leaderboards for which some models still see benefits of clones are those for which there are the most incentives for clones in the status quo mechanism: this is a result of the fact that the vote reweighting effect is larger when the number of votes per model is small.

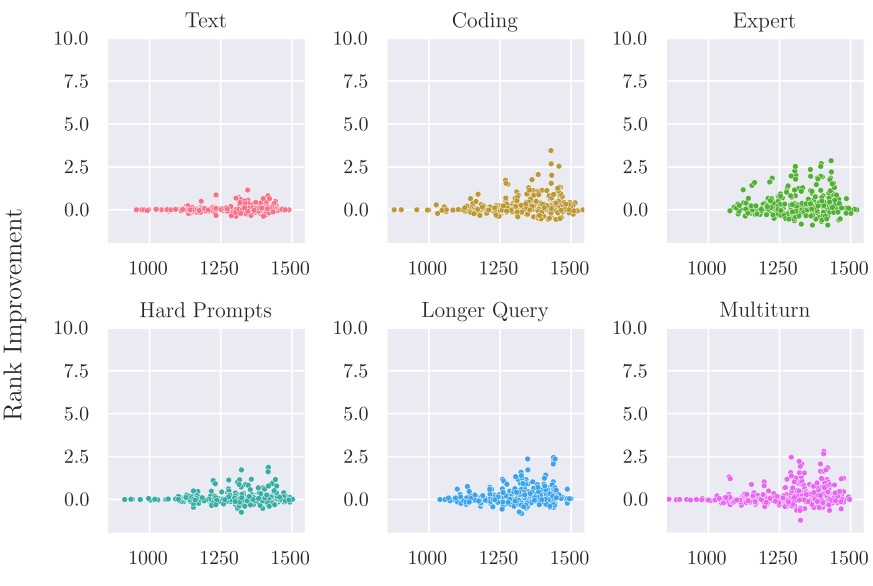

*Figure 5.* Rank difference between submitting one clone and no clones under the YRWR mechanism.

# C Preliminary Lemmata

The following theorem is rephrased from Siththaranjan et al. (2023) in our notation. We provide a proof for completeness.

**Lemma C.1** ((Siththaranjan et al., 2023), Theorem 3.1). *The rankings produced by BT-MLE and Borda counts are equivalent under equal matchup counts. Formally, let $s$ be the number of votes between each pair of candidates $i, j$. Let $\sigma^{\mathrm{BT-MLE}}$ be the ranking induced by fitting a Bradley-Terry model via MLE on $v$ as in Algorithm 1. Let $\sigma^{\mathrm{BC}}$ be the ranking induced by the Borda count on $v$. I.e., for two candidates $j, j'$*

$$j >_{\mathrm{BC}} j' \iff \sum_{\ell \in \mathcal{M}, \ell \neq j} v_{j\ell} < \sum_{\ell \in \mathcal{M}, \ell \neq j'} v_{j'\ell}.$$

*Then $\sigma^{\mathrm{BC}}(u) = \sigma^{\mathrm{BT-MLE}}(u)$ for all $u = 1, 2, \ldots$.*

As we will see in the proof, the fact that matchups are evenly distributed over pairs of models is important to the proof: if matchups are not evenly distributed, the result may not hold.

*Proof.* The claim is equivalent to showing that for any two models $j, j'$,

$$\sum_{\ell \in \mathcal{M}, \ell \neq j} v_{j\ell} < \sum_{\ell \in \mathcal{M}, \ell \neq j'} v_{j'\ell} \iff \widehat{R}_j < \widehat{R}_{j'}.$$

Now, recall that

$$\widehat{R} = \arg \max_{R \in \mathbb{R}^m} \sum_{j,j' \in \mathcal{M}; j \neq j'} v_{j>j'} \log\Big(\frac{\exp(R_j)}{\exp(R_j) + \exp(R_{j'})}\Big).$$

By concavity of the objective, it holds

$$\nabla_R \sum_{j,j' \in \mathcal{M}; j \neq j'} v_{j>j'} \log\Big(\frac{\exp(R_j)}{\exp(R_j) + \exp(R_{j'})}\Big)\Big|_{\hat{R}} = 0.$$

Also, we can rewrite each partial derivative as

$$\frac{\partial}{\partial R_j} \sum_{j,j' \in \mathcal{M}; j \neq j'} v_{j>j'} \log\Big(\frac{\exp(R_j)}{\exp(R_j) + \exp(R_{j'})}\Big)$$

$$= \sum_{j' \in \mathcal{M}, j' \neq j} \frac{\partial}{\partial R_j} v_{j > j'} \log \left( \frac{\exp(R_j)}{\exp(R_j) + \exp(R_{j'})} \right) + (s - v_{j > j'}) \log \left( \frac{\exp(R_{j'})}{\exp(R_{j'}) + \exp(R_j)} \right)$$

$$= \sum_{j' \in \mathcal{M}, j' \neq j} v_{j > j'} - \frac{s}{1 + \exp(R_{j'} - R_j)}.$$

Thus, we have

$$\frac{1}{s} \sum_{j' \in \mathcal{M}, j' \neq j} v_{j > j'} = \sum_{j' \in \mathcal{M}, j' \neq j} \frac{1}{1 + \exp(\widehat{R}_{j'} - \widehat{R}_j)}$$

by applying the fact that the partial derivative is zero at $R = \widehat{R}$. Now, the LHS is the (normalized) Borda count. Thus, for any two models $j, j'$

$$\sum_{\ell \in \mathcal{M}, \ell \neq j} v_{j\ell} < \sum_{\ell \in \mathcal{M}, \ell \neq j'} v_{j'\ell}$$

$$\iff \sum_{\ell \in \mathcal{M}, \ell \neq j} \frac{1}{1 + \exp(\widehat{R}_\ell - \widehat{R}_j)} < \sum_{\ell \in \mathcal{M}, \ell \neq j'} \frac{1}{1 + \exp(\widehat{R}_\ell - \widehat{R}_{j'})}$$

$$\iff \frac{2}{1 + \exp(R_{j'} - R_j)} - 1 + \sum_{\ell \in \mathcal{M}, \ell \neq j, \ell \neq j'} \frac{1}{1 + \exp(\widehat{R}_\ell - \widehat{R}_j)} - \frac{1}{1 + \exp(\widehat{R}_\ell - \widehat{R}_{j'})} < 0.$$

Finally, note that

$$\left( \frac{2}{1 + \exp(R_{j'} - R_j)} - 1 \right) + \sum_{\ell \in \mathcal{M}, \ell \neq j, \ell \neq j'} \left( \frac{1}{1 + \exp(\widehat{R}_\ell - \widehat{R}_j)} - \frac{1}{1 + \exp(\widehat{R}_\ell - \widehat{R}_{j'})} \right) < 0$$

$$\iff R_j - R_{j'} < 0$$

since each term inside the large parentheses is positive if $R_j > R_{j'}$ and negative if $R_j < R_{j'}$.

$\square$

## D  New Competitor Effect Analysis

In this section, we provide our workhorse stability lemma, which bounds the new competitor effect. Intuitively, it says that if the total number of votes is sufficiently large, the distributions of model scores before and after the introduction of an model cannot be too large.

Since this result may be of independent interest, we state the result for the general Bradley-Terry model and (re)introduce notation: In this section, we will consider two Bradley-Terry estimation problems:

1. an MLE $\hat{R}^m$ fit on a set of $m \geqslant 2$ candidates, and

2. an MLE $\hat{R}^{m+1}$ fit on a set of $m+1$ candidates, where the first $m$ are the same as in (1).

We'll index candidates $j = 1, 2, \ldots, m + 1$, and when fitting the MLE, we will enforce the identifiability constraint that

$$\sum_{j \in [m]} \hat{R}_j^m = \sum_{j \in [m]} \hat{R}_j^{m+1} = 0,$$

i.e., the first $m$ entries of each vector must sum to zero. We'll call these identifiable subspaces $\mathbf{1}_m^\perp$, and it will be clear from context whether we are talking about the subspace in $\mathbb{R}^m$ or $\mathbb{R}^{m+1}$, depending on whether we are working with the first or second BT estimation problems. Projecting onto $\mathbf{1}_m^\perp$ ensures that $\hat{R}^m - \hat{R}_{1:m}^{m+1} \to 0$ as $s \to \infty$ (whereas some other identifiability constraint might lead to convergence to a non-zero constant).

We will let $R \in \mathbb{R}^m$ denote the ground-truth qualities of the original $m$ models (where, without loss of generality, $\sum_{j \in [m]} R_j = 0$), and we'll write $\hat{R}_{1:m}^{m+1}$ for the entries of $\hat{R}^{m+1}$ corresponding to the first $m$ candidates. For a matrix $A$, we will similarly use index slice notation so that $A_{1:i,1:j}$ is the first $i$ rows and $j$ columns of $A$. We'll make use of Assumption 2.1 in this section, which we restate here:

**Assumption D.1.** There exists a universal constant $C$ such that, for all problem instances and $j, j' \in \mathcal{K}$, it holds $|R_j - R_{j'}| \leqslant C$.

We'll write $S = s\binom{m}{2}$ to denote the total number of pairwise comparisons, where $s$ is the number of comparisons per pair. Define $P_m$ to be the probability measure of $\hat{R}^m$ on $\mathbf{1}_m^\perp \subset \mathbb{R}^m$, and define $P_m'$ to be the probability measure of $\hat{R}_{1:m}^{m+1}$ for $\mathbf{1}_m^\perp \subset \mathbb{R}^{m+1}$.

**Lemma D.2** (New Competitor Effect). *Let $\mathcal{C}$ be the set of convex events measurable with respect to $P_m$ and $P_m'$. Then, under Assumption 2.1, there exist constants $C > 0$ and $\{C_m\}_{m \geqslant 2}$ such that for all $m \geqslant 2$ and $s \geqslant 1$,*

$$\sup_{A \in \mathcal{C}} \left| P_m(A) - P_m'(A) \right| \leqslant \frac{C}{\sqrt{m}} + \frac{C_m}{\sqrt{s}}.$$

*Proof.* At a high level, our goal will be to upper bound the convex set distance between $P_m$ and $P_m'$ by the sum of three convex set distances: the distance between $P_m$ and its Gaussian approximation, the distance between $P_m'$ and its Gaussian approximation, and the distance between the two Gaussian approximations. We then prove the Gaussian approximation error bounds in Lemma D.3 and the distance between Guassians in Lemma D.4 which immediately yield the inequality in the lemma.

Formally, let $\bar{P}_m$ denote the law of the centered and scaled estimator $\sqrt{S}(\hat{R}^m - R)$ on $\mathbf{1}_m^\perp$, and let $\bar{P}_m'$ denote the law of $\sqrt{S}(\hat{R}_{1:m}^{m+1} - R)$ on the same subspace. By $\sqrt{S}$-consistency and asymptotic normality of the MLE, there exist Normal laws $G_m, G_m'$ on $\mathbf{1}_m^\perp$ such that $\bar{P}_m \xrightarrow{d} G_m$ and $\bar{P}_m' \xrightarrow{d} G_m'$ in $S$. $G_m, G_m'$ are given by $\mathcal{N}(0, I_m^{-1}), N(0, (I_m'^{-1})_{1:m,1:m})$ where $I_m, I_m'$ are the respective Fisher information matrices projected onto $\mathbf{1}_m^\perp$.

Denote the convex-set distance as

$$d_{\mathcal{C}}(P, P') \stackrel{\text{def.}}{=} \sup_{A \in \mathcal{C}} |P(A) - P'(A)|.$$

Observe that, by the triangle inequality,

$$d_{\mathcal{C}}(\bar{P}_m, \bar{P}_m') \leqslant d_{\mathcal{C}}(\bar{P}_m, G_m) + d_{\mathcal{C}}(G_m, G_m') + d_{\mathcal{C}}(G_m', \bar{P}_m').$$

Lemma D.4 gives

$$d_{\mathcal{C}}(G_m, G_m') \leqslant \frac{C}{\sqrt{m}}$$

for a universal constant $C > 0$. Lemma D.3 yields

$$d_{\mathcal{C}}(\bar{P}_m, G_m) \leqslant \frac{C_m}{\sqrt{s}}, \quad d_{\mathcal{C}}(\bar{P}_m', G_m') \leqslant \frac{C_m'}{\sqrt{s}},$$

for constants $C_m, C_m'$ depending only on $m$. Combining these bounds gives

$$d_{\mathcal{C}}(\bar{P}_m, \bar{P}_m') \leqslant \frac{C}{\sqrt{m}} + \frac{C_m + C_m'}{\sqrt{s}}.$$

Finally, since $\hat{R}^m \mapsto \sqrt{S}(\hat{R}^m - R)$ is an invertible affine map on the identifiable subspace and $\mathcal{C}$ is the class of convex sets on that subspace, the convex-set distance is invariant under applying the same invertible affine map to both distributions. Therefore the same bound holds for the unscaled laws $P_m$ and $P_m'$, which proves the claim. $\square$

**Lemma D.3** (Gaussian approximation error for the BT–MLE). *Under Assumption 2.1, there exists a universal constant $C > 0$ such that*

$$\max \left\{ d_{\mathcal{C}}(\bar{P}_m, G_m), d_{\mathcal{C}}(\bar{P}_m', G_m') \right\} \leqslant C \frac{m^{3/4}}{\sqrt{s}} + o\left(\frac{1}{\sqrt{s}}\right).$$

**Lemma D.4** (Gaussian stability under adding one model). *Under Assumption 2.1, there exists a universal constant $C > 0$ such that*

$$d_{\mathcal{C}}(G_m, G_m') \leqslant \frac{C}{\sqrt{m}}.$$

We now proceed with the proofs for the above two lemmas. We first (re)establish general notation and basic facts we will use throughout this section. Let the win probability between candidate $i$ and $j$ be

$$p_{i>j} = \frac{\exp R_i}{\exp R_j + \exp R_i}$$

and $\hat{p}_{i>j}$ the same quantity substituting $\hat{R}$ for $R$. Let $v_{i>j} \sim \text{Bin}(s, p_{i>j})$ count the wins of model $i$ over model $j$. Let $v_{i>j}^{(k)} \sim \text{Ber}(p_{i>j})$ denote the indicator outcome of the $k$-th comparison.

Let $L_m(R)$ be the log-likelihood for $m$ candidates:

$$L_m(R) = \sum_{j,j' \in [m]; j \neq j'} v_{j>j'} \log\Big(\frac{1}{1 + \exp(R_{j'} - R_j)}\Big)$$
$$= \sum_{j,j' \in [m]; j \neq j'} v_{j>j'} \log p_{j>j'}$$

Let $\ell_m(R) = \nabla L_m(R)$ be the score function (first derivative with respect to $R$), i.e., for $j \in [m]$,

$$\ell_m(R)_j = \sum_{j' \in [m]; j' \neq j} v_{j>j'} - s p_{j>j'} \tag{D.1}$$

and let $H_m(R) = \nabla^2 L_m(R)$ be the Hessian (second derivative), i.e., for $j, j' \in [m]$,

$$H_m(R)_{j,j'} = \begin{cases} -s \sum_{\ell \in [m]; \ell \neq j} p_{j>\ell}(1 - p_{j>\ell}) & \text{if } j = j' \\ s p_{j>j'}(1 - p_{j>j'}) & \text{otherwise} \end{cases} \tag{D.2}$$
$$= -s \sum_{1 \leq j < j' \leq m} p_{j>j'}(1 - p_{j>j'})(e_j - e_{j'})(e_j - e_{j'})^\top.$$

where $e_j$ denotes the $j$th standard basis vector. Let $S = s\binom{m}{2} = O(sm^2)$ be the total number of pairwise comparisons. Let $I_m = -\mathbb{E}[H_m(R)]\big|_{\mathbf{1}_m^\perp}$ be the Fisher information matrix for the $m$-model Bradley-Terry, projected onto the rank-$(m-1)$ subspace. Let $I_{m+1}$ be the Fisher information for the $(m+1)$-model Bradley-Terry projected onto the rank-$m$ subspace.

## D.1 Proof of Lemma D.3

We will just show the bound holds for $\bar{P}_m, G_m$; the argument for $\bar{P}'_m, G'_m$ is identical. Our proof proceeds as follows:

1. We'll first write the normalized deviation of $\hat{R}^m$ from $R$ using Taylor's theorem, so that it is (up to higher-order terms) equal to the sum of independent score contributions.

2. We'll then use this linear approximation to break the convex set distance into three terms, which we can analyze separately.

3-5. Analyses of each of the terms.

**Step 1.** Applying Taylor's theorem to the score function $\ell(\cdot)$ around the MLE $\hat{R}^m$, we have

$$\ell_m(\hat{R}^m) = \ell_m(R) + H_m(\tilde{R})(\hat{R}^m - R)$$

for $\tilde{R}$ on the line segment between $R$ and $\hat{R}^m$. Also, $0 = \ell_m(\hat{R}^m)$ by the fact that the MLE is a maximum. Applying this fact and rearranging, we have

$$\hat{R}^m - R = -H_m(\tilde{R})^{-1}\ell_m(R).$$

Scaling by $\sqrt{S}$ and adding/subtracting the Fisher information

$$\sqrt{S}(\hat{R}^m - R) = \underbrace{\sqrt{S} \cdot I_m^{-1}\ell_m(R)}_{\text{Linear Approximation Term } W_m} + \underbrace{\sqrt{S}\left(-H_m(\tilde{R})^{-1} - I_m^{-1}\right)\ell_m(R)}_{\text{Remainder } r_m}$$

**Step 2.** We now break the convex set distance into three terms. First, we prove an upper bound. For $A \in \mathcal{C}$, define

$$A^t = \{x \; : \; \inf_{a \in A} \|x - a\|_2 \leqslant t\}$$

to be the $t$-neighborhood of $A$. Observe for all $t > 0$,

$$
\begin{aligned}
P(\sqrt{S}(\hat{R}^m - R) \in A) &= P(W_m + r_m \in A) && \text{(Definition of } W_m, r_m) \\
&\leqslant P(W_m \in A^t \cup \|r_m\|_2 \geqslant t) && (W_m + r_m \in A \subseteq W_m \in A^t \cup \|r_m\|_2 \geqslant t) \\
&\leqslant P(W_m \in A^t) + P(\|r_m\|_2 \geqslant t). && \text{(Union bound)}
\end{aligned}
$$

Subtracting $P(\tilde{Z} \in A^t)$ from both sides, we have

$$
\begin{aligned}
&P(\sqrt{S}(\hat{R}^m - R) \in A) - P(\tilde{Z} \in A^t) \\
&\quad \leqslant P(W_m \in A^t) - P(\tilde{Z} \in A^t) + P(\|r_m\|_2 \geqslant t) \\
&\implies P(\sqrt{S}(\hat{R}^m - R) \in A) - P(\tilde{Z} \in A) \\
&\quad \leqslant |P(W_m \in A^t) - P(\tilde{Z} \in A^t)| + P(\tilde{Z} \in A^t \backslash A) + P(\|r_m\|_2 \geqslant t) && \text{(D.3)}
\end{aligned}
$$

where the implication follows from the fact that $P(\tilde{Z} \in A^t) = P(\tilde{Z} \in A) + P(\tilde{Z} \in A^t \backslash A)$ and taking the absolute value. Note that $A^t \in \mathcal{C}$: the Minkowski sum of convex events is a convex event.

We can write the lower bound analogously. Let us overload notation and write

$$A^{-t} = \{a \in A \; : \; \inf_{x \notin A} \|x - a\|_2 \geqslant t\}.$$

Observe for all $t > 0$ that

$$
\begin{aligned}
P(\sqrt{S}(\hat{R}^m - R) \in A) &= P(W_m + r_m \in A) && \text{(Definition of } W_m, r_m) \\
&\geqslant P(W_m \in A^{-t}) - P(\|r_m\|_2 \geqslant t). && \text{(Union bound and rearranging)}
\end{aligned}
$$

Then

$$
\begin{aligned}
&P(\sqrt{S}(\hat{R}^m - R) \in A) - P(\tilde{Z} \in A^{-t}) \\
&\quad \geqslant P(W_m \in A^{-t}) - P(\tilde{Z} \in A^{-t}) - P(\|r_m\|_2 \geqslant t) \\
&\implies P(\sqrt{S}(\hat{R}^m - R) \in A) - P(\tilde{Z} \in A) \\
&\quad \geqslant -\left|P(W_m \in A^{-t}) - P(\tilde{Z} \in A^{-t})\right| - P(\tilde{Z} \in A \backslash A^{-t}) - P(\|r_m\|_2 \geqslant t). && \text{(D.4)}
\end{aligned}
$$

Similarly, note that $A^{-t} \in \mathcal{C}$.

In steps 3-5, we bound each of the terms in the right-hand side of Equation (D.3). In particular, plugging in the RHS expressions in Equations (D.5) to (D.7) yields, for $t > 0$

$$P(\sqrt{S}(\hat{R}^m - R) \in A) - P(\tilde{Z} \in A) \leqslant C\frac{m^{3/4}}{\sqrt{s}} + Cts^{-1/2}m^{-1/4} + m\exp\left(\frac{-Cst^2}{m^2}\right).$$

Choosing $t = O(m)$ yields

$$P(\sqrt{S}(\hat{R}^m - R) \in A) - P(\tilde{Z} \in A) \leqslant C\frac{m^{3/4}}{\sqrt{s}}.$$

(where as usual the constant $C$ across inequalities may change). The corresponding terms in Equation (D.4) are bounded using the same argument, and yield the same bound.

**Step 3.** We will show for a generic convex event $A$, there exists a universal constant $C$ such that

$$|P(W_m \in A) - P(\tilde{Z} \in A)| \leqslant C\frac{m^{3/4}}{\sqrt{s}}. \tag{D.5}$$

Plugging in $A^t$ or $A^{-t}$ yields our upper bound on the first terms in Equations (D.3) and (D.4), respectively.

To do this, we will first prove an approximation bound on the score function $\ell_m(R)$ and then translate it into a bound on $\sqrt{S}I_m^{-1}\ell_m(R)$.

For the bound on $\ell_m(R)$, observe that the score function at the true $R$ is a sum of $S = s\binom{m}{2}$ independent score contributions

$$\ell_m(R) = \sum_{i<j}\sum_{k=1}^{s} \psi_{ij}^{(k)} = \sum_{i<j}\sum_{k=1}^{s} (v_{i>j}^{(k)} - p_{i>j})(e_i - e_j), \qquad \mathbb{E}[\psi_{ij}^{(k)}] = 0,$$

where $e_j$ denotes the $j$-th standard basis vector. Since each term is a mean-zero independent random variable, we can apply the following theorem to bound the Gaussian approximation error on the linear term.

**Theorem D.5** (Theorem 1.1 Bentkus (2005)). *Suppose $X_1, \ldots, X_n \in \mathbb{R}^d$ are independent and $\mathbb{E}X_i = 0$ for all $i$. Let $S = \sum_i X_i$ and define $\Sigma = \mathrm{Var}(S)$. Let $Z \sim \mathcal{N}(0, \Sigma)$. There exists a universal constant $c$ such that*

$$\sup_{A\in\mathcal{C}} |P(S \in A) - P(Z \in A)| \leqslant cd^{1/4}\beta$$

*where*

$$\beta = \sum_{i=1}^{n} \|\Sigma^{-1/2}X_i\|_2^3.$$

Since $\mathrm{Cov}(\ell_m) = I_m$, the target Gaussian is $Z \sim \mathcal{N}(0, I_m)$. Plugging this into the bound from Theorem D.5 yields

$$\sup_{A\in\mathcal{C}} |P(\ell_m \in A) - P(Z \in A)| \leqslant c(m-1)^{1/4}\beta$$

where

$$\beta = \sum_{i<j}\sum_{k=1}^{s} \mathbb{E}\|I_m^{-1/2}\psi_{ij}^{(k)}\|_2^3.$$

Now it suffices to bound $\beta$. Observe that $(v_{i>j}^{(k)} - p_{i>j})) \in [-1, 1]$ so

$$\mathbb{E}\|\psi_{ij}^{(k)}\|_2^3 = \mathbb{E}\|(v_{i>j}^{(k)} - p_{i>j}))(e_i - e_j)\|_2^3$$
$$\leqslant \|e_i - e_j\|_2^3 = \sqrt{2}^3$$

By Lemma D.7, we also have

$$\|I_m^{-1/2}\|_{\mathrm{op}} \leqslant O\left(\frac{1}{\sqrt{sm}}\right)$$

which implies

$$\beta \leqslant \sum_{i<j}\sum_{k=1}^{s} O((sm)^{-3/2}) = O(sm^2)O((sm)^{-3/2}) = O(m^{1/2})O(s^{-1/2}).$$

Putting the norm upper bounds together, we obtain

$$\sup_{A\in\mathcal{C}} |P(\ell_m(R) \in A) - P(Z \in A)| \leqslant O\left(\frac{m^{3/4}}{\sqrt{s}}\right).$$

Finally, we must translate these bounds into bounds on events for $\sqrt{S}I_m^{-1}\ell_m(R)$ (rather than $\ell_m(R)$). Since convex sets are closed under linear maps, we can define the target Guassian for the scaled linear term to be $\tilde{Z} \sim \mathcal{N}(0, SI_m^{-1})$ since $\mathrm{Cov}(\sqrt{S}I_m^{-1}\ell_m(R)) = SI_m^{-1}\mathrm{Cov}(\ell_m(R))I_m^{-1} = SI_m^{-1}$ and thus have

$$\sup_{A\in\mathcal{C}} |P(W_m \in A) - P(\tilde{Z} \in A)| = \sup_{A\in\mathcal{C}} |P(\ell_m(R) \in A) - P(Z \in A)| \leqslant O\left(\frac{m^{3/4}}{\sqrt{s}}\right).$$

**Step 4.** We will show

$$P(\tilde{Z} \in A^t \backslash A) \leqslant Cts^{-1/2}m^{-1/4}. \tag{D.6}$$

To do so, we apply the following theorem:

**Theorem D.6** (Nazarov (2003))**.** *There exist universal constants $0 < C_1 < C_2 \in \mathbb{R}$ such that, for any mean-zero multivariate Gaussian measure $F$ on $\mathbb{R}^d$ with variance matrix $W$, it holds*

$$C_1\sqrt{\|W\|_{\mathrm{F}}} \leqslant \sup_{A \in \mathcal{C}, t > 0} \frac{F(A \backslash A^t)}{t} \leqslant C_2\sqrt{\|W\|_{\mathrm{F}}}.$$

In particular, we have $\|\Sigma\|_{\mathrm{F}} \leqslant \sqrt{m}\|\Sigma\|_{\mathrm{op}} \leqslant \sqrt{m}O(1/(sm))$ where the last inequality follows from Lemma D.7. Thus, there is a universal constant $C$ such that

$$\sup_{Q \in \mathcal{C}, h > 0} \frac{P(\tilde{Z} \in Q^h \backslash Q)}{h} \leqslant Cs^{-1/2}m^{-1/4}$$

Thus, for fixed $t$, we have

$$\begin{aligned}
P(\tilde{Z} \in A^t \backslash A) &= t \cdot \frac{P(\tilde{Z} \in A^t \backslash A)}{t} \\
&\leqslant t \cdot \sup_{Q \in \mathcal{C}, h > 0} \frac{P(\tilde{Z} \in Q^h \backslash Q)}{h} \\
&\leqslant Cts^{-1/2}m^{-1/4}.
\end{aligned}$$

**Step 5.** Finally, we establish

$$P(\|r_m\|_2 \geqslant t) \leqslant m \exp\left(\frac{-Cst^2}{m^2}\right) \tag{D.7}$$

Notice

$$\begin{aligned}
\|r_m\|_2 &= \left\|\sqrt{S}\left(-H_m(\tilde{R})^{-1} - I_m^{-1}\right)\ell_m(R)\right\|_2 \\
&\leqslant \sqrt{S}\left\|-H_m(\tilde{R})^{-1} - I_m^{-1}\right\|_{\mathrm{op}}\|\ell_m(R)\|_2 \\
&\leqslant \sqrt{S}\left(\left\|H_m(\tilde{R})^{-1}\right\|_{\mathrm{op}} + \left\|I_m^{-1}\right\|_{\mathrm{op}}\right)\|\ell_m(R)\|_2
\end{aligned}$$

Now, we bound each of these terms. From Lemma D.7, we have $\|I_m^{-1}\|_{\mathrm{op}} = O((sm)^{-1})$. Moreover, using the same proof as for Lemma D.7, under the assumption that $\max_{j,j'}|\hat{R}_j - \hat{R}_{j'}|$ is bounded and the fact that $\tilde{R}$ is a convex combination of $R, \hat{R}$, it holds $\|H_m(\tilde{R})^{-1}\|_{\mathrm{op}} = O((sm)^{-1})$. Thus,

$$\|r_m\|_2 \leqslant O(s^{-1})\|\ell_m(R)\|_2.$$

Moreover, from Lemma D.8, for all $t$, we have

$$P(\|\ell_m(R)\| \geqslant t) \leqslant m \exp\left(-\frac{t^2}{3sm^2}\right)$$

Plugging in $O(s \cdot t)$ for $t$, we have

$$P(\|r_m\| \geqslant t) = m \exp\left(\frac{-C \cdot s \cdot t^2}{m^2}\right).$$

$\square$

## D.2 Proof of Lemma D.4

Let $\Sigma = I_m^{-1}$ and $\Sigma' = (I_{m+1}^{-1})_{1:m,1:m}$. By asymptotic normality (as $s \to \infty$) of the MLE, we can write out $G_m, G'_m$ explicitly

$$\sqrt{S}(\hat{R}^m - R) \to \mathcal{N}(0, S\Sigma)$$

$$\sqrt{S}(\hat{R}_{1:m}^{m+1} - R) \to \mathcal{N}(0, S\Sigma')$$

Since convex-set distance is invariant under scaling by a constant, we define

$$\tilde{G}_m = \mathcal{N}(0, \Sigma)$$

$$\tilde{G}'_m = \mathcal{N}(0, \Sigma')$$

Then

$$
\begin{aligned}
d_{\mathcal{C}}(G_m, G'_m) &= d_{\mathcal{C}}(\tilde{G}_m, \tilde{G}'_m) \\
&\leqslant d_{\mathrm{TV}}(\tilde{G}_m, \tilde{G}'_m) \\
&\leqslant \sqrt{\frac{1}{2}\mathrm{KL}(\tilde{G}_m \| \tilde{G}'_m)} \\
&= \frac{1}{2}\left(\mathrm{tr}(\Sigma'^{-1}\Sigma) - (m-1) + \log\frac{\det\Sigma'}{\det\Sigma}\right).
\end{aligned}
$$

(D.8)

(D.9)

where the last line is the formula for the KL-divergence between two centered multivariate normal distributions. Thus, showing that the convex-set distance between $G_m, G'_m$ is small can be done in terms of $\Sigma, \Sigma'$ by showing Equation (D.9) is small.

To bound Equation (D.9), we will proceed with the following steps:

1. We will decompose $\Sigma' = (\Sigma + K)^{-1}$ for some matrix $K$ determined by the change in log-likelihood due to the additional model.

2. We will establish $\|K\|_{\mathrm{op}} \leqslant s/4$ and use this to upper bound the expression for the KL-divergence between multivariate normals.

**Step 1.** With an additional model added, we can write the log-likelihood as

$$L_{m+1}(R_{1:m}, R_{m+1}) \overset{\mathrm{def.}}{=} \underbrace{L_m(R_{1:m})}_{\text{original comparisons}} + \underbrace{L_{\text{new}}(R_{1:m}, R_{m+1})}_{\text{comparisons involving the new model}},$$

where $L_m(R_{1:m})$ is the log-likelihood for comparisons between the original $m$ models, and

$$L_{\text{new}}(R_{1:m}, R_{m+1}) \overset{\mathrm{def.}}{=} \sum_{j \in [m]} v_{j>m+1} \log p_{j>m+1} + (s - v_{j>m+1}) \log(1 - p_{j>m+1}).$$

By linearity of expectations and gradients, we then have,

$$I_{m+1} = -\mathbb{E}[\nabla_R^2 L_m(R_{1:m})] - \mathbb{E}[\nabla_R^2 L_{\text{new}}(R_{1:m}, R_{m+1})].$$

We will further simplify these expressions by writing a block decomposition for each term. For the first term, $L_m$ depends only on $R_{1:m}$, so

$$-\mathbb{E}[\nabla^2 L_m(R)] = \begin{pmatrix} I_m & 0 \\ 0 & 0 \end{pmatrix}.$$

The second term can be written as

$$-\mathbb{E}[\nabla_R^2 L_{\text{new}}(R_{1:m}, R_{m+1})] = \begin{pmatrix} I_{1:m,1:m}^{\text{new}} & I_{1:m,(m+1)}^{\text{new}} \\ I_{(m+1),1:m}^{\text{new}} & I_{(m+1),(m+1)}^{\text{new}} \end{pmatrix}$$

where

$$I^{\text{new}}_{1:m,1:m} \overset{\text{def.}}{=} -\mathbb{E}\big[\nabla^2_{R_{1:m}} L_{\text{new}}(R)\big],$$

$$I^{\text{new}}_{1:m,(m+1)} \overset{\text{def.}}{=} -\mathbb{E}\left[\nabla_{R_{1:m}} \frac{\partial}{\partial R_{m+1}} L_{\text{new}}(R)\right], \text{ and}$$

$$I^{\text{new}}_{(m+1),(m+1)} \overset{\text{def.}}{=} -\mathbb{E}\left[\frac{\partial^2}{\partial^2 R_{m+1}} L_{\text{new}}(R)\right].$$

Thus the Fisher information for the log-likelihood with the additional model is

$$I_{m+1} = \begin{pmatrix} I_m + I^{\text{new}}_{1:m,1:m} & I^{\text{new}}_{1:m,(m+1)} \\ I^{\text{new}}_{(m+1),1:m} & I^{\text{new}}_{(m+1),(m+1)} \end{pmatrix}$$

Applying the block inversion formula yields

$$\Sigma' = (I^{-1}_{m+1})_{1:m,1:m} = \left(I_m + I^{\text{new}}_{1:m,1:m} - I^{\text{new}}_{1:m,(m+1)}(I^{\text{new}}_{(m+1),(m+1)})^{-1} I^{\text{new}}_{(m+1),1:m}\right)^{-1}$$

Finally, define

$$K \overset{\text{def.}}{=} I^{\text{new}}_{1:m,1:m} - I^{\text{new}}_{1:m,(m+1)}(I^{\text{new}}_{(m+1),(m+1)})^{-1} I^{\text{new}}_{(m+1),1:m}$$

so

$$\Sigma' = (I_m + K)^{-1}.$$

**Step 2.** Since $K$ is the Schur complement of $I^{\text{new}}_{(m+1),(m+1)} \geq 0$, it holds $K \geq 0$. We write out the partial derivatives under Bradley-Terry. Let $d_i = s \cdot p_{i,m+1}(1 - p_{i,m+1}) \leqslant s/4$ and $D = \texttt{diag}(d)$. Then,

$$-\nabla^2_{R_{1:m}} L_{\text{new}}(R) = s \cdot D$$

$$-\nabla_{R_{1:m}} \frac{\partial}{\partial R_{m+1}} L_{\text{new}}(R) = -s \cdot d, \text{ and}$$

$$-\frac{\partial^2}{\partial^2 R_{m+1}} L_{\text{new}}(R) = s\mathbf{1}^\top d.$$

Thus, taking expectations (all terms are deterministic), we have

$$I^{\text{new}}_{1:m,1:m} = s \cdot D$$
$$I^{\text{new}}_{1:m,(m+1)} = I^{\text{new}\top}_{(m+1),1:m} = -s \cdot d$$
$$I^{\text{new}}_{(m+1),(m+1)} = s \cdot \mathbf{1}^\top d$$

Then we can rewrite $K$ as

$$K = I^{\text{new}}_{1:m,1:m} - I^{\text{new}}_{1:m,(m+1)}(I^{\text{new}}_{(m+1),(m+1)})^{-1} I^{\text{new}}_{(m+1),1:m}$$

$$= D - \frac{dd^\top}{\mathbf{1}^\top d}.$$

Moreover, $D - K \geq 0$, since for any vector $x$, it holds $(x^\top d)(d^\top x) = (x^\top d)^2 \geqslant 0$ and $\mathbf{1}^\top d > 0$. Also, let $I$ be the identity matrix and $u = D^{1/2}\mathbf{1}/\sqrt{\mathbf{1}^\top d}$,

$$D - \frac{dd^\top}{\mathbf{1}^\top d} = D^{1/2}\left(I - uu^\top\right) D^{1/2}$$

So $\|K\|_{\text{op}} \leqslant \|D^{1/2}\|^2_{\text{op}} \|I - uu^\top\|_{\text{op}}$. Finally,

$$\left\|I - uu^\top\right\|_{\text{op}} \leqslant 1$$

since $\|u\|_2 = 1$ so the non-zero eigenvalue of $uu^\top$ is 1, which means that the eigenvalues of $I - uu^\top$ are all 1 except one which is zero. Thus,

$$\|K\|_{\mathrm{op}} \leqslant \|D\|_{\mathrm{op}} \leqslant \frac{s}{4}. \tag{D.10}$$

With these facts in hand, we now proceed to upper bound Equation (D.9). Denote the relative perturbation matrix

$$A = \Sigma^{1/2} K \Sigma^{1/2} = I_m^{-1/2} K I_m^{-1/2}.$$

Observe that $A \succeq 0$ since for any vector $x$,

$$x^T \Sigma^{1/2} K \Sigma^{1/2} x = (x^T \Sigma^{1/2}) K (\Sigma^{1/2} x) \geqslant 0$$

by positive semidefiniteness of $K$. Moreover,

$$\|A\|_{\mathrm{op}} \leqslant \|\Sigma^{1/2}\|_{\mathrm{op}}^2 \|K\|_{\mathrm{op}} \leqslant O\left(\frac{1}{sm}\right) \frac{s}{4} \leqslant O\left(\frac{1}{m}\right)$$

under Assumption 2.1 by applying Lemma D.7 and Equation (D.10).

We first compute the trace term in Equation (D.9). Observe,

$$\begin{aligned}
\mathrm{tr}(\Sigma'^{-1}\Sigma) &= \mathrm{tr}\big((I_m + K)I_m^{-1}\big) \\
&= \mathrm{tr}(\mathbf{I}) + \mathrm{tr}(KI_m^{-1}) \\
&= (m-1) + \mathrm{tr}(I_m^{-1/2} K I_m^{-1/2}) \\
&= (m-1) + \mathrm{tr}(A)
\end{aligned}$$

where the last equality applies the cyclic property of trace. Next, for the determinant term in Equation (D.9), observe that

$$\Sigma' = (I_m + K)^{-1} = \Sigma^{1/2}(\mathbf{I} + A)^{-1}\Sigma^{1/2}$$

which implies

$$\det(\Sigma') = \det(\Sigma)\det\big((\mathbf{I}) + A)^{-1}\big).$$

Therefore,

$$\log\frac{\det\Sigma'}{\det\Sigma} = \log\det\big((\mathbf{I} + A)^{-1}\big) = -\log\det(\mathbf{I} + A)$$

Substituting into the KL divergence formula yields

$$\mathrm{KL}(\tilde{G}_m \| \tilde{G}'_m) = \frac{1}{2}\Big(\mathrm{tr}(A) - \log\det(\mathbf{I} + A)\Big)$$

Let $\lambda_1, \ldots, \lambda_{m-1}$ denote the eigenvalues of $A$, then

$$\mathrm{tr}(A) = \sum_i \lambda_i, \quad \det(\mathbf{I} + A) = \prod_i (1 + \lambda_i),$$

and

$$\mathrm{KL}(\tilde{G}_m \| \tilde{G}'_m) = \frac{1}{2}\sum_{i=1}^{m-1}\big(\lambda_i - \log(1 + \lambda_i)\big).$$

For all $\lambda \geqslant 0$, it is true that

$$\log(1 + \lambda) \geqslant \lambda - \frac{\lambda^2}{2}$$

and it follows that

$$\lambda_i - \log(1 + \lambda_i) \leqslant \lambda_i - \left(\lambda_i - \frac{\lambda_i^2}{2}\right) = \frac{\lambda_i^2}{2}$$

Summing over all eigenvalues gives:

$$\mathrm{KL}(\tilde{G}_m \| \tilde{G}'_m) \leqslant \frac{1}{4} \sum_{i=1}^{m-1} \lambda_i^2 \leqslant \frac{1}{4} \|A\|_F^2 \leqslant \frac{(m-1)}{4} \|A\|_{\mathrm{op}}^2 \leqslant O(1/m)$$

Thus,

$$d_{\mathcal{C}}(G_m, G'_m) \leqslant O(m^{-1/2})$$

as desired. $\qquad\square$

**Lemma D.7.** *Under Assumption 2.1, there exists a universal constant $\eta \in (0, 1/2)$ such that*

$$\|I_m\|_{\mathrm{op}} \geqslant \eta(1-\eta)sm \tag{D.11}$$

*and hence $\|I_m^{-1}\|_{\mathrm{op}} \leqslant (\eta(1-\eta)sm)^{-1}$.*

*Proof of Lemma D.7.* Note that under Assumption 2.1, there exists a universal constant $\eta = 1/(1 + \exp(C)) \in (0, 1/2)$ such that $p_{i>j} \in [\eta, 1-\eta]$ for all $i \neq j$, and hence $p_{i>j}(1 - p_{i>j}) \geqslant \eta(1-\eta) > 0$. Therefore, for any $x \in \mathbf{1}^{\perp}$ such that $\|x\|_2 = 1$,

$$x^{\top} I_m x = s \sum_{i<j} p_{i>j}(1 - p_{i>j})(x_i - x_j)^2 \geqslant s\eta(1-\eta) \sum_{i<j}(x_i - x_j)^2.$$

Since

$$\sum_{i<j}(x_i - x_j)^2 = \frac{1}{2} \sum_{i,j} x_i^2 + x_j^2 - 2x_i x_j$$

$$= m \sum_i x_i^2 - \left(\sum_i x_i\right)^2 = m$$

on $\mathbf{1}^{\perp}$, we obtain $x^{\top} I_m x \geqslant s\eta(1-\eta)m$, which implies $\|I_m\|_{\mathrm{op}} \geqslant s\eta(1-\eta)m$ and

$$\|I_m^{-1}\|_{\mathrm{op}} \leqslant \frac{1}{s\eta(1-\eta)m}.$$

$\qquad\square$

**Lemma D.8.** *Under Assumption 2.1, there exists a universal constant such that, for all $\varepsilon > 0$ and $s \geqslant 3\log(2m/\varepsilon)/\eta$, it holds with probability at least $1 - \varepsilon$ that,*

$$\|\ell_m(R)\|_2 \leqslant C\sqrt{sm^2 \log(m/\varepsilon)}$$

*Proof.* Recall,

$$\|\ell_m(R)\|_2^2 = \sum_{j \in [m]} \left(\sum_{j' \in [m]; j' \neq j} v_{j>j'} - sp_{j>j'}\right)^2.$$

Now, applying a Chernoff bound, we have with probability $1 - \varepsilon/m$

$$\left|\sum_{j' \in [m]; j' \neq j} v_{j>j'} - sp_{j>j'}\right| \leqslant \sqrt{3s \log(2m/\varepsilon) \sum_{j' \in [m]; j \neq j'} p_{j>j'}}$$

$$\leqslant \sqrt{3sm \log(2m/\varepsilon)}.$$

Thus, with probability at least $1 - \varepsilon$,

$$\sum_{j \in [m]} \left(\sum_{j' \in [m]; j' \neq j} v_{j>j'} - sp_{j>j'}\right)^2 \leqslant \sum_{j \in [m]} 3sm \log(2m/\varepsilon)$$

$$= 3sm^2 \log(2m/\varepsilon).$$

$\qquad\square$

# E    Proof of Theorem 3.2

We first restate the result for reference.

**Theorem 3.2** (Clone-nonrobustness of the status quo mechanism). *For all constants $\varepsilon, \delta > 0$, there exists $s_0, m_0$ such that for all $s \geqslant s_0, m \geqslant m_0$, the following holds. For any producer $i$, any strategy profiles $z$ and any $(\varepsilon, \delta)$-competitive model $j$, producer $i$ would benefit from submitting an additional copy of $j$. Formally, let $z' = (z_{i,j} + 1, z_{-i,j})$. Then*

$$\mathbb{E}_{\sigma \sim \Sigma(\mathrm{SQ}(\mathcal{M}(z')))}[u_i(\sigma)] > \mathbb{E}_{\sigma \sim \Sigma(\mathrm{SQ}(\mathcal{M}(z)))}[u_i(\sigma)].$$

In our proof, we'll call the set of candidates induced by $z$ the "original candidates" and $j^{(z_{i,j}+1)}$ the "additional clone". Similarly, we'll call the vote distributions induced by $z$, $z'$ respectively as the "original distribution" and the "additional clone distribution". At a high level, our proof will proceed as follows:

1. We'll observe that the win probability of a producer with an additional clone is equal to the probability that some model by the producer is ranked above all of the original candidates.

2. Next, we'll argue that the event that the additional clone ranks above all original candidates and the event that any of the original candidates by the same producer rank above the original candidates are anticorrelated. This implies the probability (with respect to the additional clone distribution) that a model by the producer with the additional clone is ranked first is no less than the probability the additional clone is ranked first plus the probability one of the original candidates by the producer is ranked first minus the product of these two probabilities.

3. We then translate these two probabilities into events that are measurable with respect to the original distribution, and apply Lemma D.2 to establish that the probabilities of these two events may differ from their probabilities in the original distribution by at most $O(1/\sqrt{s})$.

4. These facts together imply that if Definition 3.1 is satisfied, the producer's win probability with a clone is greater than without it, which completes the proof.

Without loss of generality, suppose producer $i = 1$'s model $j = 1$ satisfies Definition 3.1. Let $w = z_{1,1} + 1$. Thus, the clone is indexed $1^{(w)}$. Define $\mathcal{M}_{-1} = \mathcal{M}(z) \backslash \mathcal{M}_1(z_1)$ to be all candidates but those submitted by producer 1. For a model $j$ by producer 1, a leaderboard $L$, and a random ranking $\sigma$, define the event

$$A_j(L) = \left\{\sigma^{-1}(j) < \sigma^{-1}(\ell) \ \ \forall \ell \in \mathcal{M}_{-1} \cap L\right\}.$$

That is, $A_j(L)$ is the event that a model $j$ ranks above all those by other producers $\mathcal{M}_{-1}$ in the leaderboard $L$. We will overload notation and write, for a set of candidates $S$,

$$A_S(L) = \bigcup_{j \in S} A_j(L).$$

For $S \subseteq \mathcal{M}_1(z_1)$ and $\sigma \sim \Sigma(\mathrm{SQ}, z)$, note that

$$A_S = \left\{\sigma_L(1) \in S\right\}.$$

That is, if any model by producer 1 ranks above all candidates by other producers in $L$ under actions $z$, a model by producer 1 must be ranked first in $L$. Moreover, if model $1^{(w)} \in S$, $S \subseteq \mathcal{M}_i$, and $\sigma \sim \Sigma(\mathrm{SQ}, z')$, then $A_S = \{\sigma_L(1) \in S\}$: if $S$ contains the new clone and is a subset of producer 1s candidates, one model in $S$ must be ranked first in $L$ for producers' actions $z'$.

Now, for the expectation and probabilities taken with respect to $\sigma \sim \Sigma(\mathrm{SQ}, z')$, observe

$$\mathbb{E}[u_1(\sigma)] = \sum_{L \in \mathcal{L}} \nu_1(L) \Pr(\sigma_L(1) \in \mathcal{M}_1(z_1')) \qquad \text{(Definition of utility.)}$$

Now, notice

$$\Pr(A_{\mathcal{M}_1(z_1')}(L)) = \Pr(A_{\mathcal{M}_1(z_1)}(L) \cup A_{1^{(w)}}(L)) \qquad\qquad (\mathcal{M}_1(z_1') = \mathcal{M}_1(z_1) \cup \{1^{(w)}\})$$

$$= \Pr(A_{\mathcal{M}_1(z_1)}(L)) + \Pr(A_{1^{(w)}}(L)) - \Pr(A_{\mathcal{M}_1(z_1')}(L) \cap A_{1^{(w)}}(L)) \quad \text{(Inclusion-exclusion formula.)}$$

$$\geqslant \Pr(A_{\mathcal{M}_1(z_1)}(L)) + \Pr(A_{1^{(w)}}(L)) - \Pr(A_{\mathcal{M}_1(z_1')}(L))\Pr(A_{1^{(w)}}(L)) \qquad\qquad \text{(Lemma E.1)}$$

$$= \Pr(A_{\mathcal{M}_1(z_1)}(L)) + \Pr(A_{1^{(1)}}(L)) - \Pr(A_{\mathcal{M}_1(z_1')}(L))\Pr(A_{1^{(1)}}(L)). \qquad (\sigma_L^{-1}(1^{(1)}) \overset{d}{=} \sigma_L^{-1}(1^{(w)}))$$

Plugging the last expression into the sum above, we have

$$\mathbb{E}[u_1(\sigma)] = \sum_{L \in \mathcal{L}} \nu_1(L) \left( \Pr(A_{\mathcal{M}_1(z_1)}(L)) + \Pr(A_{1^{(1)}}(L)) - \Pr(A_{\mathcal{M}_1(z_1')}(L))\Pr(A_{1^{(1)}}(L)) \right)$$

Now, observe that the events $A_{\mathcal{M}_1(z)}(L)$ and $A_{1^{(1)}}(L)$ are measurable with respect to $\sigma \sim \Sigma(\text{SQ}, z)$ (the original distribution). Thus, applying Lemma D.2, for all $\nu > 0$, there exists $s_0, m_0$ such that for $s \geqslant s_0, m \geqslant m_0$,

$$\Pr_{\sigma \sim \Sigma(\text{SQ},z')}(A_{\mathcal{M}_1(z)}(L)) \geqslant \Pr_{\sigma \sim \Sigma(\text{SQ},z)}(A_{\mathcal{M}_1(z)}(L)) - \nu, \text{ and}$$

$$\Pr_{\sigma \sim \Sigma(\text{SQ},z')}(A_{1^{(1)}}(L)) \geqslant \Pr_{\sigma \sim \Sigma(\text{SQ},z)}(A_{1^{(1)}}(L)) - \nu.$$

Combining these with the expression above, we have

$$\mathbb{E}_{\sigma \sim \Sigma(\text{SQ},z')}[u_1(\sigma)] \geqslant \sum_{L \in \mathcal{L}} \nu_1(L) \Bigg( \Pr_{\sigma \sim \Sigma(\text{SQ},z)}(A_{\mathcal{M}_1(z)}(L)) + \Pr_{\sigma \sim \Sigma(\text{SQ},z)}(A_{1^{(1)}}(L))$$

$$- \Pr_{\sigma \sim \Sigma(\text{SQ},z)}(A_{\mathcal{M}_1(z)}(L)) \cdot \Pr_{\sigma \sim \Sigma(\text{SQ},z)}(A_{1^{(1)}}(L)) \Bigg) - \nu$$

$$\geqslant \sum_{L \in \mathcal{L}} \nu_1(L) \Pr_{\sigma \sim \Sigma(\text{SQ},z)}(A_{\mathcal{M}_1(z)}(L))$$

$$+ \sum_{L \in \mathcal{L}} \nu_1(L) \delta \big(1 - \Pr_{\sigma \sim \Sigma(\text{SQ},z)}(A_{\mathcal{M}_1(z)}(L))\big) - \nu \qquad \text{(Definition 3.1, first inequality)}$$

$$\geqslant \sum_{L \in \mathcal{L}} \nu_1(L) \Pr_{\sigma \sim \Sigma(\text{SQ},z)}(A_{\mathcal{M}_1(z)}(L)) + \sum_{L \in \mathcal{L}} \nu_1(L)\delta^2 - \nu \qquad \text{(Definition 3.1, second inequality)}$$

$$\geqslant \sum_{L \in \mathcal{L}} \nu_1(L) \Pr_{\sigma \sim \Sigma(\text{SQ},z)}(A_{\mathcal{M}_1(z)}(L)) + \varepsilon\delta^2 - \nu \qquad \text{(Definition 3.1, $\varepsilon$ condition)}$$

Finally, as long as we set $\nu < \varepsilon\delta^2$, we have

$$\sum_{L \in \mathcal{L}} \nu_1(L) \Pr_{\sigma \sim \Sigma(\text{SQ},z)}(A_{\mathcal{M}_1(z)}(L)) + \varepsilon\delta^2 - \nu$$

$$\geqslant \sum_{L \in \mathcal{L}} \nu_1(L) \Pr_{\sigma \sim \Sigma(\text{SQ},z)}(A_{\mathcal{M}_1(z)}(L))$$

$$= \mathbb{E}_{\sigma \sim \Sigma(\text{SQ},z)}[u_1(\sigma)]$$

where the last line is by definition. $\qquad\square$

**Lemma E.1.** *It holds*

$$\Pr(A_{M_1(z_1)}(L) \cap A_{1^{(w)}})(L) \leqslant \Pr(A_{M_1(z_1)}(L))\Pr(A_{1^{(w)}}(L)).$$

*Proof of Lemma E.1.* From Lemma C.1, since matchup counts are allocated evenly across pairs, the rankings induced by Bradley-Terry fit with MLE and Borda counts are equivalent. Thus, if the set of candidates submitted to the mechanism is $\mathcal{M}(z')$, we have that

$$A_j(L) = \left\{ \sum_{\ell \in \mathcal{M}(z')} v_{j > \ell} > \sum_{\ell \in \mathcal{M}(z')} v_{j' > \ell} \ \ \forall j' \in \mathcal{M}_{-1} \cap L \right\},$$

and similarly for $A_S(L)$. Now, let $F = \{v_{1(w)>j}\}_{j \in \mathcal{M}_1}$ be all vote counts between the additional clone and producer 1's original candidates, $\mathcal{M}_1$. Let $F^C = v \backslash F$ be all other vote counts, keeping one copy of each independent vote count and excluding all $\{v_{j>1(w)}\}_{j \in \mathcal{M}_1}$. (I.e., if $v_{j>j'} \in F^C$ then $v_{j'>j} \notin F^C$, since $v_{j>j'} + v_{j'>j} = s$.) We will argue that

$$\Pr(A_{M_1(z_1)}(L) \cap A_{1(w)}(L) \mid F^C) \leqslant \Pr(A_{M_1(z_1)}(L) \mid F^{(C)}) \Pr(A_{1(w)}(L) \mid F^{(C)}). \tag{E.1}$$

This implies the result since the conditional probabilities for all $F^{(C)}$ imply the unconditional ones (by taking expectations with respect to $F^{(C)}$ for the left- and right-hand sides of the equation). To show Equation (E.1), we will apply the Harris inequality.

Note that the measure induced by $F$ (conditional on $F^C$) is a product measure, since vote counts between different pairs of candidates are independent. Thus, it is sufficient to show that $A_{M_1(z_1)}$ is a decreasing event and $A_{1(w)}$ is an increasing event. Now, writing each event in terms of the Borda count, we can rewrite $A_{1(w)}(L)$ as

$$\sum_{\ell \in \mathcal{M}_i(z_i'), \ell \neq 1(w)} v_{1(w)>\ell} > \sum_{\ell \in \mathcal{M}(z'), \ell \neq j} v_{j>\ell} - \sum_{\ell \in \mathcal{M}_{-1}} v_{1(w)>\ell}, \quad \forall j \in \mathcal{M}_{-1} \cap L.$$

Note that the left-hand side sum is over elements in $F$ and the right-hand side sums are over elements determined by $F^C$. Now, by inspecting the left-hand side, note that the event $A_{1(w)} \mid F^C$ is increasing: if the inequality is satisfied and we increase an entry of $v_{1(w)>\ell}$, the inequality is still satisfied.

Similarly, we can write $A_{M_1(z_1)}(L)$ as

$$\exists j \in \mathcal{M}_1(z) \cap L \text{ s.t. } v_{j>1(w)} > \sum_{\ell \in \mathcal{M}(z'), \ell \neq j'} v_{j'>\ell} - \sum_{\ell \in \mathcal{M}(z), \ell \neq j} v_{j>\ell}, \quad \forall j' \in \mathcal{M}_{-1} \cap L.$$

Now, since $v_{j>1(w)} + v_{1(w)>j} = s$, we can equivalently write $A_{M_1(z_1)}(L)$ as

$$\exists j \in \mathcal{M}_1(z) \cap L \text{ s.t. } v_{1(w)>j} < s - \sum_{\ell \in \mathcal{M}(z'), \ell \neq j'} v_{j'>\ell} + \sum_{\ell \in \mathcal{M}(z), \ell \neq j} v_{j>\ell}, \quad \forall j' \in \mathcal{M}_{-1} \cap L.$$

Again, the LHS contains terms in $F$ and the RHS contains terms determined by $F^C$. Thus, the event $A_{M_1(z_1)}$ is decreasing: If the inequality is satisfied for some $j \in M_1(z) \cap L$ and we decrease $v_{1(w)>j'}$ for some $j' \in M_1(z) \cap L$, the inequality is still satisfied. Thus, by the Harris inequality, Equation (E.1) is satisfied and the result holds. $\qquad \square$

## F    Proof of Theorem 4.1

We first restate the result:

**Theorem 4.1** (Approximate cloneproofness). *For all $\varepsilon > 0$, there exists $s_0, m_0$ such that for all $s \geqslant s_0$ and $m \geqslant m_0$, the following holds. Fix any $\pi, z$, and let $z' = (\mathbf{1}, z_{-i})$ be the profile where $i$ instead plays one copy of each model. Then*

$$\mathbb{E}_{\sigma \sim \Sigma(\text{YRWR}(\mathcal{M}(z'), \pi)}[u_i(\sigma)])$$
$$\geqslant \mathbb{E}_{\sigma \sim \Sigma(\text{YRWR}(\mathcal{M}(z), \pi))}[u_i(\sigma)] - \varepsilon.$$

In our proof, we'll apply the following two lemmas. The first says that all producers approximately prefer to submit at least 1 copy of each model.

**Lemma F.1.** *For all $\varepsilon > 0$, there exists $s_0, m_0$ such that for all $s \geqslant s_0$ and $m \geqslant m_0$, the following holds. Consider an action vector $z$ where $z_{i,j} = 0$ for some producer $i$ and model $j$. Let $z' = (1, z_{-(i,j)})$ be the action where $i$ submits one copy of $j$. Let $\pi'$ be the same as $\pi$ on all pairs ranked by $\pi$ and let the newly submitted model be producer-ranked last. Then*

$$\mathbb{E}_{\sigma \sim \Sigma(\text{YRWR}, z', \pi')}[u_i(\sigma)] \geqslant \mathbb{E}_{\sigma \sim \Sigma(\text{YRWR}, z, \pi)}[u_i(\sigma)] - \varepsilon.$$

The second says that, for any producer $i$ action $z_i$ where there is some $z_{ij} > 1$, it holds the producer approximately prefers to submit one copy of $j$.

**Lemma F.2.** *For all $\varepsilon > 0$, there exists $s_0, m_0$ such that for all $s \geqslant s_0$ and $m \geqslant m_0$, the following holds. Consider an action vector $z$ where $z_{i,j} > 1$ for some producer $i$ and model $j$. Let $z' = (1, z_{-(i,j)})$ be the action where $i$ submits one copy of $j$. Let $\pi'$ be equal to the $\pi$ when dropping $j^{(2)}, \ldots, j^{(z_{ij})}$, keeping the ordering of remaining candidates the same. Then*

$$\mathbb{E}_{\sigma \sim \Sigma(\text{YRWR}, z', \pi')}[u_i(\sigma)] \geqslant \mathbb{E}_{\sigma \sim \Sigma(\text{YRWR}, z, \pi)}[u_i(\sigma)] - \varepsilon.$$

We will show that together, these lemmas imply the theorem: Intuitively, for any producer $i$ action $z_i$, we can make a series of $\varepsilon$-approximately utility-improving changes to the action such that we end up with action $\mathbf{1}$. And since each producer only has at most a constant number of distinct candidates by assumption, there are only a constant number of such changes. Setting $\varepsilon$ appropriately then yields the theorem (i.e., if $W$ is the maximum number of distinct models, setting $\varepsilon$ in the lemma equal to $\varepsilon/W$ for $\varepsilon$ in the theorem).

Formally, let $z^{(0)} = z$ and $\pi^{(0)} = \pi$. For $u \in k_i$, let $z^{(u)} = (1, z_{-(i,u)}^{(u-1)})$ be the action that sets the first $u$ entries of $z$ to 1. Note that $z_i^{(m_i)} = \mathbf{1}$ so $z^{(m_i)} = z'$. Similarly, define $\pi_i^{(u)}$ to be the ranking achieved by altering $\pi_i^{(u-1)}$ by

1. appending entry $u$ to the end of the ranking if $z_{i,u} = 0$,

2. dropping $j^{(2)}, \ldots, j^{(z_{i,u})}$ from the ranking if $z_{i,u} > 1$, and

3. keeping the ranking as is if $z_{i,u} = 1$.

Note that $\pi_i^{(m_i)} = \mathbf{1}$, so $\pi^{(m_i)} = \pi'$. By telescoping, note that

$$\mathbb{E}_{\sigma \sim \Sigma(\text{YRWR}, z', \pi')}[u_i(\sigma)] - \mathbb{E}_{\sigma \sim \Sigma(\text{YRWR}, z, \pi)}[u_i(\sigma)]$$
$$= \sum_{u=1}^{m_i} \mathbb{E}_{\sigma \sim \Sigma(\text{YRWR}, z^{(u)}, \pi^{(u)})}[u_i(\sigma)] - \mathbb{E}_{\sigma \sim \Sigma(\text{YRWR}, z^{(u-1)}, \pi^{(u-1)})}[u_i(\sigma)]. \tag{F.1}$$

At each step we apply either Lemma F.1 if $z_{i,j} = 0$ or Lemma F.2 if $z_{i,j} > 1$ and note that $z^{(u-1)} = z^{(u)}$ and $\pi^{(u-1)} = \pi^{(u)}$ otherwise, so the expectations are equal in that case. Thus, each term in the sum on the second line is no less than $-\varepsilon/W$ and so the sum is no less than $-\varepsilon$. Rearranging the left-hand side of Equation (F.1) yields the inequality in the theorem.

Without loss of generality, in the next proofs of the next two lemmas, let the focal producer in each lemma be indexed 1 so that we may use $i$ for generic producers.

*Proof of Lemma F.1.* Let $\sigma \sim \Sigma(\text{YRWR}, z, \pi)$ and $\sigma' \sim \Sigma(\text{YRWR}, z', \pi')$. Then

$$\mathbb{E}_{\sigma'}[u_1(\sigma')] - \mathbb{E}_{\sigma}[u_1(\sigma)]$$
$$= \sum_{L \in \mathcal{L}} \nu_1(L) \left( \Pr_{\sigma'}[\sigma'(1) \in \mathcal{M}_1(z_1')] - \Pr_{\sigma}[\sigma(1) \in \mathcal{M}_1(z_1)] \right) \qquad \text{(Definition of } u_1)$$

$$= \sum_{L \in \mathcal{L}} \nu_1(L) \left( \sum_{\ell \in \mathcal{M}_1(z_1')} \Pr_{\sigma'}[\sigma'(1) = \ell] - \sum_{\ell \in \mathcal{M}_1(z_1)} \Pr_{\sigma}[\sigma(1) = \ell] \right) \quad \text{(Probabilities that a model ranks first are disjoint.)}$$
$$= \sum_{L \in \mathcal{L}} \nu_1(L) \Pr_{\sigma'}[\sigma'(1) = j] + \sum_{L \in \mathcal{L}} \nu_1(L) \sum_{\ell \in \mathcal{M}_1(z_1)} \Pr_{\sigma'}[\sigma'(1) = \ell] - \Pr_{\sigma}[\sigma(1) = \ell] \qquad (\mathcal{M}_1(z_1') = \mathcal{M}_1(z_1) \cup \{j\}.)$$

Let $\bar{m}$ be an upper bound on $\max_i |\mathcal{M}_i(z_i)|$ (since we have assumed that no producer has more than a constant number of models). Finally, the first term is trivially nonnegative and each term in the second sum is no less than $-\varepsilon/\bar{m}$ by Lemma D.2 (choosing $\varepsilon$ in Lemma D.2 to be $\varepsilon/\bar{m}$). Thus, each inner sum is no less than $-\varepsilon$ and by assumption on $\nu$ summing to no more than 1, the outer sum must be no less than $-\varepsilon$. $\qquad \square$

*Proof of Lemma F.2.* Let $\sigma \sim \Sigma(\text{YRWR}, z, \pi)$ and $\sigma' \sim \Sigma(\text{YRWR}, z', \pi')$. Then,

$$\mathbb{E}_{\sigma'}[u_1(\sigma')] - \mathbb{E}_{\sigma}[u_1(\sigma)]$$

$$= \sum_{L \in \mathcal{L}} \nu_1(L) \left( \sum_{u=2}^{z_{ij}} \Pr_{\sigma'}(\sigma'_L(1) = j^{(u)}) + \sum_{\ell \in \mathcal{M}_1(z_1)} \Pr_{\sigma'}(\sigma'_L(1) = \ell) - \Pr_{\sigma}(\sigma_L(1) = \ell) \right),$$

similar to Lemma F.2. Now, notice that $\sum_{u=2}^{z_{ij}} \Pr_{\sigma'}(\sigma'_L(1) = j^{(u)}) = 0$ since WLOG we assumed that $\pi(j^{(1)}) < \pi(j^{(u)})$ for $u > 1$, so the mechanism ensures $\sigma(j^{(1)}) < \sigma(j^{(u)})$ for $u > 1$. Moreover, each term in the second RHS inner sum is no less than $-\varepsilon/W$, since we removed at most $\bar{m} - 1$ candidates and so can apply Lemma D.2 (using parameter $\varepsilon/(\bar{m}W)$) $\bar{m}$ times for each term. Finally, there are $W$ terms in the second RHS sum by assumption, so the whole sum is no less than $-\varepsilon$.

$\square$

# G  Additional proofs

*Proof of Proposition 4.2.* To show the key inequality above, we will specifically prove the following chain of inequalities:

$$\|\check{R}_s - R\|_\infty = \|T_\pi(\hat{R}_s) - T_\pi(R)\|_\infty \leqslant \|\hat{R}_s - R\|_\infty$$

where the first step is by definition and the fact that when $\pi$ is truthful, $T_\pi(R) = R$ (the rewards are unchanged by the correction). The second inequality is by the general lipschitzness of the correction, which we will prove now:

Fix $R', R'' \in \mathbb{R}^m$, fix a producer $i$ and $j \in M_i$. Let $S_{i,j} := \{\pi_i(k) : k \leqslant \pi_i^{-1}(j)\}$ be the set of all candidates $i$ ranked in $\pi_i$ as better than $j$. Then

$$|(T_\pi(R'))_j - (T_\pi(R''))_j| = \left| \min_{j' \in S_{i,j}} R'_{j'} - \min_{j'' \in S_{i,j}} R''_{j''} \right| \leqslant \max_{j' \in S_{i,j}} |R'_{j'} - R''_{j'}| \leqslant \|R' - R''\|_\infty.$$

The first inequality is by the fact that if the mins are very far apart, then either those mins correspond to the same $j'$ (and it holds with equality), or they correspond to different $j', j''$ and if so, the distance between the mins is a lower bound on the distance between the respective rewards for at least one of these candidates. The second inequality is just by the definition of the $\ell_\infty$ norm (lefthand side is just max over a subset of candidates).

Then, it follows that

$$\|T_\pi(R') - T_\pi(R'')\|_\infty = \max_j |(T_\pi(R'))_j - (T_\pi(R''))_j| \leqslant \|R' - R''\|_\infty. \qquad \square$$

*Proof Corollary 4.3.* By Proposition 4.2,
$$\|\check{R}_s - R\|_\infty \leqslant \|\hat{R}_s - R\|_\infty.$$

This implies the claim because if for all random realizations of $\hat{R}_s$ this is true, then for any $M$ and any $s$,

$$\Pr(\|\check{R}_s - R\|_\infty > M/\sqrt{s}) \leqslant \Pr(\|\hat{R}_s - R\|_\infty > M/\sqrt{s}),$$

And $\sqrt{s}$ consistency is shown. Correctness under $\gamma$-separated true rewards is a direct implication of the first part, since the probability any pair of models is misranked goes to zero as $s \to \infty$. $\square$

*Proof of Proposition 4.5.* We begin by assuming that all candidates have distinct qualities. Under this assumption, there exists some $\gamma > 0$ such that $|R_j - R_{j'}| > \gamma$ for all $j, j'$. Because MLE estimates for Bradley-Terry concentrate around their true values, $\hat{R}_j - \hat{R}_{j'}$ concentrates around $R_j - R_{j'}$. Therefore, if $R_j > R_{j'}$, then

$$\Pr[\hat{R}_j - \hat{R}_{j'} < 0] \xrightarrow[s \uparrow \infty]{} 0.$$

Consider any $j, j' \in \mathcal{K}_i$ such that $R_j > R_{j'}$. Suppose they are adjacently ranked in $\pi_i$, but in the incorrect order (i.e., $\cdots > j' > j > \ldots$). Swapping their ranks (call this $\pi'_i$) can only reduce producer $i$'s utility in the case where $\hat{R}_{j'} > \hat{R}_j$. As shown above, the probability that this occurs goes to 0 as $s \uparrow \infty$. Therefore, for any $\varepsilon$, there exists sufficiently large $s$ such that $\Pr[\hat{R}_j - \hat{R}_{j'} < 0] < \varepsilon$.

Swapping from $\pi_i$ to $\pi_i'$ can only weakly increase $\breve{R}_j$ and weakly decrease $\breve{R}_{j'}$. We ignore the effect of increasing $\breve{R}_j$, since it can only increase utility. Decreasing $\breve{R}_{j'}$ can reduce utility because $j'$ may lose a leaderboard it would have otherwise won, and occurs if and only if $\hat{R}_{j'} > \hat{R}_j$. Formally,

$$\mathbb{E}_{\sigma \sim \Sigma(\text{YRWR}(\mathcal{M}(z), \pi_i'))}[u_i(\sigma)] - \mathbb{E}_{\sigma \sim \Sigma(\text{YRWR}(\mathcal{M}(z), \pi_i'))}[u_i(\sigma)]$$

$$\leqslant \mathbb{E}_{\sigma \sim \Sigma(\text{YRWR}(\mathcal{M}(z), \pi_i'))}[u_i(\sigma) \cdot \mathbf{1}(\hat{R}_{j'} > \hat{R}_j)] - \mathbb{E}_{\sigma \sim \Sigma(\text{YRWR}(\mathcal{M}(z), \pi_i'))}[u_i(\sigma) \cdot \mathbf{1}(\hat{R}_{j'} > \hat{R}_j)]$$

$$\leqslant \Pr[\hat{R}_{j'} > \hat{R}_j]$$

$$\leqslant \varepsilon$$

for sufficiently large $s$. Applying this argument inductively (we need at most $\binom{k_i}{2}$ swaps) shows that the true ranking $\pi_i^*$ is an $\varepsilon$-approximate dominant strategy for sufficiently large $s$.

Finally, we can lift the assumption that all qualities are distinct by allowing that either ranking of two candidates with identical qualities is considered truthful. Our argument applies to all pairs of candidates with distinct scores, which yields the result. □

*Proof of Corollary A.1.* Let $\mathscr{A}$ be the event that the confidence intervals cover $R$. Now, on $\mathscr{A}$, the analysis in the proof of Theorem 4.1 holds: Since the confidence intervals cover $R$, clones must have overlapping confidence intervals. Thus, the isotonic score correction will be applied to all clones and the arguments for the approximate utility improvement induced by removing clones apply as is (conditioning on $\mathscr{A}$). On $\mathscr{A}^C$, since utilities are bounded in $[0, 1]$, the change in utility can be at most one. And since the confidence intervals are simultaneously valid, $\mathscr{A}^C$ holds with probability at most $\varepsilon$. Therefore:

$$\mathbb{E}_{\sigma \sim \Sigma(\text{UA-YRWR}(\mathcal{M}(z'), \pi, \alpha)}[u_i(\sigma)]) - \mathbb{E}_{\sigma \sim \Sigma(\text{UA-YRWR}(\mathcal{M}(z), \pi, \alpha))}[u_i(\sigma)]$$

$$= P(\mathscr{A})(\mathbb{E}_{\sigma \sim \Sigma(\text{UA-YRWR}(\mathcal{M}(z'), \pi, \alpha)}[u_i(\sigma) \mid \mathscr{A}]) - \mathbb{E}_{\sigma \sim \Sigma(\text{UA-YRWR}(\mathcal{M}(z), \pi, \alpha))}[u_i(\sigma) \mid \mathscr{A}])$$

$$\quad + (1 - P(\mathscr{A}))(\mathbb{E}_{\sigma \sim \Sigma(\text{UA-YRWR}(\mathcal{M}(z'), \pi, \alpha)}[u_i(\sigma) \mid \mathscr{A}^C]) - \mathbb{E}_{\sigma \sim \Sigma(\text{UA-YRWR}(\mathcal{M}(z), \pi, \alpha))}[u_i(\sigma) \mid \mathscr{A}^C])$$

$$\geqslant 1 \cdot (-\varepsilon) + \alpha \cdot (-1)$$

□

*Proof of Proposition A.2.* For $\sqrt{s}$-consistency, note that since the confidence intervals are $\sqrt{s}$ consistent and include $\hat{R}$, for all $j \in \mathcal{M}$

$$\breve{R}_j^{\text{UA}} - \hat{R}_j = O_P(s^{-1/2}).$$

Moreover, by the MLE theorem, $\hat{R}_j - R_j = O_P(s^{-1/2})$. Thus,

$$\breve{R}_j^{\text{UA}} - R_j = (\breve{R}_j^{\text{UA}} - \hat{R}_j) - (R_j - \hat{R}_j) = O_P(s^{-1/2}).$$

Correctness is a direct implication of the first part, since the probability any pair of models is misranked goes to zero as $s \to \infty$. □

