# OpenReview forum: "Strategic Candidacy in Generative AI Arenas"
_ICML.cc/2026/Conference — ICML 2026 regular_

### Official Review · Reviewer_kGTb · 2026-03-11

**Soundness:** 4
**Presentation:** 3
**Significance:** 3
**Originality:** 3
**Overall Recommendation:** 5
**Confidence:** 3

**Summary:**

The submission proposes a "strategy-robust" ranking method for preference-ranked models in AI leaderboards. Under a specific formulation of the strategic submission problem, it shows that the new method is approximately able to remove the benefit of multi-submission, and improve the overall ranking accuracy. In addition to formal proofs of the setup, the submission validates the approach in a simulated experiments generated from a snapshot of the real LMArena leaderboard.

**Compliance With Llm Reviewing Policy:**

Affirmed.

**Final Justification:**

I think the rebuttal addressed some of my concerns and questions around presentation and the paper is solid overall. I am not raising my score higher because of the narrowness of the setup, as noted in my rebuttal response.

**Key Questions For Authors:**

See my questions above in the discussion of weaknesses, key ones being:
1. Does the theory apply in (what I think is) the more realistic setup of utility associated with leaderboard position and or "being in the competitive class"?
2. What is doing the work in the proofs, especially as it comes to the priority ordering (does it have to be part of the producer's strategy?) and the ranks estimated from BT?
3. Are there no other relevant baselines to include?

**Limitations:**

On the whole yes (though as noted above I think the choice of utility function is doing a lot of work here that is not fully unpacked).

**Strengths And Weaknesses:**

# Strengths

The work is relevant to a practical problem in AI leaderboards, and is set up in a generally relevant way, which is supported by experiments alongside formal results, which also appear sound (though I have not reviewed them in detail). From a presentation perspective, the framing is clear and I appreciate the informal intuitions accompanying the formal definitions and the proof setup. I also think the new competitor effect wasn't obvious to me, and identifying the opposition between it and the lottery ticket effect is interesting. While the mechanism proposed is straightforward and quite similar to what is already used at ICML, I buy the argument that the setup here is novel.

# Weaknesses

1. First, a minor presentation nit: I needed to draw out a decision tree to understand the interaction between leaderboard ranking and producer priority. Specifically:

a. If a new model is the new best, and producer would've ranked it highest priority, they should just submit it and nothing else.

b. If the new model is the new best, and producer would've not ranked it highest priority, its score would be reduced to the score of the top priority model. Thus, no incentive to submit it.

c. If the new model is not the new best, but producer would've ranked it highest, then the score of the producer's prior best would be *reduced* to match it. Thus, no incentive to submit.

d. If the new model is not the new best and producer would have not ranked it highest, it doesn't improve their best leaderboard position. Thus, no incentive to submit.

Perhaps a table or flow chart to this effect would help with clarity.

2. Next, regarding the setup: it's surprising that nonzero utility is assigned to a producer only for winning a leaderboard. I think at least from PR verbiage, producers care about having a "competitive" model (defined e.g. as a model statistically indistinguishable from the top, or within some epsilon from the top, rather than strictly the top). For example in conventional discussion Gemini, Claude, and GPT are considered "on par with each other" and other producers compete to be "in the frontier class" rather than strictly best. On the other hand, if one model was far and away the best, it becomes the "frontier class". I think in such a setup, the lottery ticket effect should intuitively diminish (because all clones would be within statistical closeness to each other), and the new competitor effect likewise. This does diminish somewhat the argument that the setup matches real incentives on real leaderboards. On the other hand, I think there are other setups where strict rankings matter (e.g. Kaggle / data science competition leaderboards), though in those cases rankings are not usually derived from BT. This is discussed briefly in the conclusion but could be surfaced earlier and possibly discussed in more detail.

3. Regarding the proofs and definitions: the paper would benefit from being more explicit of what is doing the work in the proofs. There's some good examples of this (e.g. noting early in section 4 that any deterministic prioritization / ordering of a producer's models is sufficient to eliminate the lottery ticket effect). But it could be improved: for example, does the ranking need to be producer-determined, or merely fixed? As far as I can tell if within-producer ordering is unknown to the producer but fixed, then perhaps theorem 4.1 still holds (they still are best off submitting only their best model, because otherwise the monotone regression effect on arbitrary rankings might reduce the score of their top model), but proposition 4.2 may no longer hold. Similarly I wonder if YRWR would still work in a setup where utility is proportional to leaderboard rank (by e.g. defining a "pseudo-leaderboard" only including positions up to rank N whose utility is less than the leaderboard including positions up to rank N+1). On the other hand, I suspect it would not work in the "competitive for its class" setup. Also on the note of "doing the work", is the BT model required anywhere for the proofs? I'm not sure if it is, but maybe I am missing something.

4. Regarding the empirical results: it's a bit disappointing not to see any baselines besides SQ (for one, the isotonic regression approach of Su et al. might apply here even if the formal setup is a bit different). I also had a curiosity about reporting "rank improvement" -- is it the case that cloning under YRWR might also reduce the rank of a producer's best model (by submitting a model the producer ranks highest but is not highest)? Showing this might be additional evidence that YRWR disincentivizes cloning.

Finally, a nit: the classic "Garden of Forking Paths" note from Gelman and Loken (https://sites.stat.columbia.edu/gelman/research/unpublished/p_hacking.pdf) is an early discussion of strategic effects in analysis that seem to me to be in the same line of thinking as e.g. the Speiss 2025 citation. I will leave it to the authors to determine if it is relevant enough to cite.

---

> ### Author Rebuttal · Authors · 2026-03-27
>
> Thank you for your thoughtful review! In particular, we appreciated that you found the question we study interesting and practical, the paper and framing well-written, and the provided intuition/analysis non-obvious/insightful. We will add discussion corresponding to your points in the manuscript, which will strengthen our work.
>
> Weaknesses:
> 1. Thanks for this suggestion. We agree with your description of the interaction between leaderboard ranking and producer ranking, with the caveat that everything you say holds *for a single leaderboard analyzed in isolation*. When models show up in different leaderboards, the story may become more complicated because the producer may have to weigh the benefits of winning one leaderboard against the possible costs of preventing a higher priority model from winning on another leaderboard.
> 2. Thanks for this thoughtful point. Perhaps another way to formalize your intuition is that "BT scores matter more than rankings"? I.e., if there are several models all with similar, top BT scores (a frontier class), these models all derive some utility. This is plausible, and indeed if this were the case there would be less of an incentive (but still non-zero incentive) to produce clones. However, our intuition is still that being ranked first matters a lot to companies: Even within the frontier class, model producers gain publicity (and subscribers who consult the leaderboard) from being the top ranked.
> 3. We will be more explicit about the key ideas in the proofs. To your concrete points:
>
> On who creates the “producer ranking”: You are right that what we call the "producer ranking" need not be created by the producers themselves --- as long as the "producer ranking" is fixed in advance of voting, any ranking eliminates the benefits of clones. However, we believe producers are best-equipped to rank their models (and we wanted to analyze their incentives in submitting rankings), so we focus on producer rankings created by producers. Similarly, you are correct that if the producer ranking (regardless of whether created  by the producer or some other process) yields an overall incorrect ranking, the accuracy of the ranking could go down.
>
> On other utility models: We don't think that classical position-based utility models (beyond winner take-all) make sense in our setting. For example, how do we compare rankings of different lengths? If there is utility for a producer even for ranking near-last, they might be incentivised to submit clones just to increase the length of the list regardless of their competitiveness, and this didn't seem as interesting of an incentive problem to solve (although the results would be the same under the status quo: incentives for submitting clones).
>
> 4. Regarding the use of the BT model in proofs: The BT model is used extensively in the proofs. For example, our workhorse lemma (Appendix C) argues that the BT fitted scores are approximately normal even in finite samples and that the voter reallocation effect does not change these normal distributions very much. If the data generating process were different, a different analysis would need to be done.
>
> On other baselines: We focus on the status quo (SQ) mechanism because it is actually implemented in the most popular arenas, and so is the most relevant baseline. The Su et al. mechanism is conceptually very similar to ours except that it would essentially average scores rather than taking the minimum. For our setting, where there are strategic candidacy concerns, taking the minimum solves the incentives problem better than averaging would (i.e., averaging still creates small selection effects, but taking the minimum precisely eliminates them). By contrast, in the Su et al. peer review setting, there is no strategic candidacy (the set of candidates is fixed in advance), so averaging scores via isotonic regression serves a different purpose (denoising). We aren't aware of other reasonable baselines for our setting.
>
> In response to your point about “rank improvement”: Indeed, if the producer ranks some other model higher than their best model, the other model could bring down the rank of their best model. However, if the producer ranks their best model first, no YRWR score correction will be applied.
>
> Questions:
>
> 1. One would need to conduct a different analysis (although parts of proofs could be reused) if utility were based on, e.g., the gap in scores rather than ranking. The incentives for cloning would persist but be attenuated. We suspect the YRWR mechanism would still disincentivise clones. The key feature of our utility function is that it is increasing in the maximum performance of a model, and as long as this assumption is met, there are likely other utility functions (like a “frontier class” utility function) for which our results would still hold.
>
> 2. See discussion of the producer ranking above.
>
> 3. See discussion of baselines above.

---

> > ### Author Rebuttal · Reviewer_kGTb · 2026-03-31
> >
> > I appreciate the authors rebuttals and promise to clarify and strengthen the paper. I remain positive on it and look forward to seeing it at the conference, though I think that the relative narrowness of the setup (BT only, top leaderboard rank only, SQ baseline only, etc) keep me from raising my score further.

---

### Official Review · Reviewer_Q1mr · 2026-03-12

**Soundness:** 3
**Presentation:** 3
**Significance:** 2
**Originality:** 2
**Overall Recommendation:** 4
**Confidence:** 4

**Summary:**

The paper studies an interesting problem in AI arenas: clone-robustness. It first shows that under the current mechanism, a producer can benefit substantially from submitting cloned models. It then proposes a new mechanism, called YRWR, which uses each producer’s ranking over their own models to estimate model quality. The authors prove that this mechanism is approximately clone-robust.

**Compliance With Llm Reviewing Policy:**

Affirmed.

**Final Justification:**

My question has been resolved, and I still view the paper positively overall. It provides a good starting point for studying strategic behavior in arena-style LLM evaluation systems, though the modeling choices make the scope a bit narrower than I would have liked.

**Key Questions For Authors:**

1. While Theorem 3.2 shows that cloning is always beneficial, the model does not appear to include any submission cost. Does this mean that, in the current framework, the optimal strategy is simply to submit as many copies as possible?

2. Following up on the previous question: Example 3.3 illustrates the benefit of cloning for a single model. However, if all producers were to adopt the same strategy, the resulting winning probabilities might remain unchanged. In this case, what kind of Nash equilibrium, if any, should we expect? More generally, does the model imply an arms-race dynamic in which all producers prefer to submit as many copies as possible?

3. It would also be helpful if the authors could comment on the extent to which the results generalize beyond the homogeneous-preference setting, in particular to settings with heterogeneous voter preferences.

**Limitations:**

1. The key assumption (Definition 3.1) appears to be a fairly strong interiority assumption. It excludes both weak models that can never win and dominant producers who are guaranteed to win with some submitted model. It would be helpful to discuss how essential this assumption is for the result.

2. Proposition 4.2 (on YRWR is always accuracy-improving) compared the performance of YRWR with the correct ranking $\pi^*$ with the performance of SQ, which feels like an unfair comparison that works in favor of the YRWR mechanism.

**Strengths And Weaknesses:**

Soundness: The submission is overall technically sound, and the main claims are well supported by the theoretical analysis. The proofs appear correct, although the results rely on somewhat strong assumptions (see the limitations below).

Presentation: The paper is clearly written, well-organized, and easy to follow overall. It also does a good job of positioning the work in the context of AI arenas.

Significance: The paper tackles an interesting and timely question in LLM evaluation with AI Arenas. It studies the limitations of the current mechanism and proposes an alternative mechanism to address them.

Originality: The paper contributes to the literature on clone-robustness in Bradley–Terry-style models, which is an interesting direction. At the same time, the problem considered here seems more tractable than the fully general setting, since under homogeneous preferences the randomness arises entirely from noise, so overall the problem is easier to analyze.

---

> ### Author Rebuttal · Authors · 2026-03-27
>
> Thank you for your thoughtful review! We appreciated that you find the research questions interesting and timely, the results sound, and the paper/framing well-written. We address your questions below.
>
> Responses to key questions:
> 1. You are correct that there are no submission costs in the model. We made this choice because it is free to submit new models to popular platforms like LMArena. In the model, the optimal strategy would indeed be to submit as many models as possible. In reality, there may be implicit costs, caps on the number of models a producer can submit or reputational risks from having been discovered submitting an enormous number of models to the platform. To avoid bloat in our model (and to avoid having to assume what form these costs or risks might take), we do not model these costs and instead analyze the benefits of the addition of a single model.
> 2. Yes, our results suggest an arms race where all producers prefer to submit as many copies as possible (up to the new competitor effect). Win probabilities might not change much in an equilibrium (or they might get closer to uniform if the number of votes per pair does not stay constant when large numbers of models are added to the system) when producers submit as many copies as possible, but the quality of the ranking might go down due to the additional ranking noise.
> 3. We focus on the homogeneous preference case because it illustrates that there is an incentives problem *even if the statistical model used for inference is well-specified*. In a more general model, perhaps other incentive problems could also emerge.  Additionally, we aren't sure that rankings "make sense" under heterogeneous preferences, since the ground truth / ideal ranking isn’t straightforward to define in this context. I.e., how should a platform rank 3 models (A, B, C) with 3 equal sub-populations of voters where sub-population 1 prefers A to B to C, sub-population 2 prefers B to C to A, and sub-population 3 prefers C to A to B? We think the right answer to this is to create separate arenas for each subpopulation and to analyze them separately. This is practically challenging if the sub-populations are hard to identify, but is very practical (and arguably already implemented in LMArena) if the sub-populations are easy to identify, like those that use LLMs for coding, those that use them for math, and those that use them for creative writing, for example. Finally, we remark that we suspect our results may still largely hold under heterogeneous preferences: The basic intuition with the problems of strategic candidacy and the solution that YRWR provides are the same: Model producers can benefit from statistical noise from voters (including under heterogeneous preferences) and the YRWR mechanism eliminates “selection-on-winners” effects.
>
> Limitations:
> 1. We note that the key assumption is not an assumption on all models, just on models that will benefit from submitting clones to the mechanism. (Thus, our results do not preclude the existence of weak models or dominant producers.) The assumption is very necessary: If a model has no chance of winning, there is no point in cloning it, because the clone will also have essentially no chance of winning. If a model producer will always win, there is no benefit to cloning because it will always win already -- it can't improve its utility any more!
> 2. Our intuition is that, in most contexts, model producers, via private or internal testing, should have a strong sense of the internal rankings of their models before release, and so should be able to submit the correct ranking. However, of course it is possible that producers have uncertainty about the quality of their models, in which case they may misreport rankings. In our updated manuscript, we will emphasize that accuracy improvements are conditional on truthfulness, and will not persist under extreme producer misranking. In Figure 3b, we show that under moderate misranking, YRWR will improve rank quality.

---

> > ### Author Rebuttal · Reviewer_Q1mr · 2026-04-02
> >
> > My question has been resolved, and I still view the paper positively overall. It provides a good starting point for studying strategic behavior in arena-style LLM evaluation systems, though the modeling choices make the scope a bit narrower than I would have liked.

---

### Official Review · Reviewer_dWUP · 2026-03-13

**Soundness:** 2
**Presentation:** 2
**Significance:** 3
**Originality:** 3
**Overall Recommendation:** 4
**Confidence:** 1

**Summary:**

The paper studies the mechanism design problem in generative AI arenas where producers can game the system by submitting multiple copies of the same model. The authors provide an analysis showing the flaw of an existing mechanism and propose a new one (with an additional variation) that is clone-robust. The analysis and evaluations are extensive.

**Compliance With Llm Reviewing Policy:**

Affirmed.

**Final Justification:**

The rebuttal addressed my concerns regarding several clarity issues. I retain my overall positive evaluation of the paper.

The paper is generally well written, with extensive and in-depth discussions surrounding their proposed mechanism. Although the main theoretical results are not easy for non-expert readers to grasp, the problem setup is clearly introduced and the discussions of their solution is thorough.

**Key Questions For Authors:**

Can the authors provide more explanation and discussion to the clarity issues that I outlined above?

**Limitations:**

Yes.

**Strengths And Weaknesses:**

Soundness: The paper is sound overall, but some parts are under-explained and therefore difficult to judge. To avoid redundancy, I list these under the presentation aspect.

Presentation: The paper is well written in general, but there are several unclear technical details:
- The two probabilities in Definition 3.1 are both lower bounded by $\delta$ and given that the two associated events are not independent, it is unclear how that would restrict the value of $\delta$. In particular, if the probability of getting at least 1 win is smaller than $1-\delta$, it is unclear how this is connected to the probability of model $j$ winning being greater than $\delta$.
- Although the authors discuss Theorem 3.2 in the paper, it is still unclear what roles do $(\epsilon,\delta)$ play in that result, as these variables do not appear in the result.
- The number $m$ does not seem to be defined. It is mentioned in Figure 2.
- Similarly, it is unclear what roles do $(\epsilon, s, s_0)$ play in the result of Proposition 4.5.

Significance: The paper addresses an important problem in practice, where producers can submit multiple copies of the same model. The authors also provide broader implications and insights from their results.

Originality: The work is original with sufficient discussion of related work and existing mechanisms.

---

> ### Author Rebuttal · Authors · 2026-03-27
>
> Thank you for your thoughtful review! We are grateful you found the question we study interesting, the results sound and the paper well-written. We will add clarifications corresponding to your questions in the manuscript and try to address them below.
>
> Presentation questions:
> 1. We use a single constant for conciseness, but this is without loss of generality. We could have also defined a separate constant for each inequality $\delta_1$, $\delta_2$. To translate back to our definition 3.1, it is sufficient to define $\delta = \min (\delta_1, \delta_2)$. Thus, if it is possible to compute each probability separately, we can compute a single constant that holds for both by taking the minimum. There is no “restriction on the value of $\delta$” other than this.
> 2. Thanks for this comment. We will clarify the role of these constants in the formal result and surrounding discussion. The parameter $\varepsilon$ represents how important the set of leaderboards must be, and $\delta$ represents the minimum probability that model $j$ wins on these leaderboards. Together, these parameters determine when it is beneficial for a model producer to clone. Intuitively, the smaller $\varepsilon$ and $\delta$ are, the weaker the benefits to cloning may be --- smaller $\epsilon$ means that the cloned model sits on less important (to the model producer) leaderboards and smaller $\delta$ means that the lottery ticket effect of an additional copy of the ($\epsilon$, $\delta$)-competitive model are smaller. These parameters cannot be zero or else there would be no benefit to cloning.
> 3. The variable $m$ (sometimes $m(z)$) is defined around line 116 in the manuscript. It is the total number of models submitted to the mechanism (for a given strategy profile $z$). We leave the argument $z$ implicit when it is clear from context. We'll remind readers of notation in the caption of Figure 2.
> 4. The parameter $\varepsilon$ in Proposition 4.5 is an approximation error parameter which controls "how much worse" reporting the true producer ranking can be relative to some other ranking. For a given $\varepsilon$, $s_0$ tells us the minimum per-pair vote count for which, for all sample sizes at least $s_0$, reporting the ranking truthfully cannot be more than $\varepsilon$ worse than the utility derived from any other producer ranking. The idea is that one can choose $\varepsilon$ to be arbitrarily small, so that truthful reporting yields essentially the same utility as some other report, but smaller $\varepsilon$ means that the sample size must be larger. Intuitively, finite sample effects may (slightly) distort the incentives so that in very small sample sizes, it is slightly worse to report truthfully than some other report.

---

> > ### Author Rebuttal · Reviewer_dWUP · 2026-04-04
> >
> > Thank the authors for your clarifications. As the authors mentioned, it would be great if these clarifications can be incorporated into the revision to make their theoretical results more accessible to non-expert readers.

---

### Decision · Program_Chairs · 2026-04-30

**Decision:**

Accept (regular)

**Comment:**

The paper is overall technically sound, with its main claims supported by theoretical analysis and consistent experimental evidence, though some results rely on relatively strong assumptions. While the work is well written and clearly structured, several technical details are under-specified or difficult to interpret, which makes parts of the analysis harder to follow. Despite these issues, the paper addresses a timely and practical problem in AI evaluation systems, particularly challenges arising from model duplication in leaderboard-style settings. It offers useful conceptual insights into different effects influencing ranking behavior and proposes a simple but meaningful mechanism that is largely aligned with existing approaches. Overall, the work is original and relevant, but would benefit from improved clarity in the formal presentation. I recommend a weak accept.